# Current Data and New Insights into the Genetic Factors of Atherogenic Dyslipidemia Associated with Metabolic Syndrome

**DOI:** 10.3390/diagnostics13142348

**Published:** 2023-07-12

**Authors:** Lăcramioara Ionela Butnariu, Eusebiu Vlad Gorduza, Elena Țarcă, Monica-Cristina Pânzaru, Setalia Popa, Simona Stoleriu, Vasile Valeriu Lupu, Ancuta Lupu, Elena Cojocaru, Laura Mihaela Trandafir, Ștefana Maria Moisă, Andreea Florea, Laura Stătescu, Minerva Codruța Bădescu

**Affiliations:** 1Department of Medical Genetics, Faculty of Medicine, “Grigore T. Popa” University of Medicine and Pharmacy, 700115 Iasi, Romania; ionela.butnariu@umfiasi.ro (L.I.B.); vgord@mail.com (E.V.G.); andreeaflorea97@gmail.com (A.F.); 2Department of Surgery II—Pediatric Surgery, “Grigore T. Popa” University of Medicine and Pharmacy, 700115 Iasi, Romania; 3Odontology-Periodontology, Fixed Prosthesis Department, Faculty of Dental Medicine, “Grigore T. Popa” University of Medicine and Pharmacy, 700115 Iasi, Romania; stoleriu_simona@yahoo.com; 4Department of Pediatrics, Faculty of Medicine, “Grigore T. Popa” University of Medicine and Pharmacy, 700115 Iasi, Romania; valeriulupu@yahoo.com (V.V.L.); anca_ign@yahoo.com (A.L.); laura.trandafir@umfiasi.ro (L.M.T.); stefana-maria.moisa@umfiasi.ro (Ș.M.M.); 5Department of Morphofunctional Sciences I, “Grigore T. Popa” University of Medicine and Pharmacy, 700115 Iasi, Romania; elena2.cojocaru@umfiasi.ro; 6Medical III Department, Faculty of Medicine, “Grigore T. Popa” University of Medicine and Pharmacy, 700115 Iasi, Romania; laura.statescu@umfiasi.ro; 7III Internal Medicine Clinic, “St. Spiridon” County Emergency Clinical Hospital, 1 Independence Boulevard, 700111 Iasi, Romania; minerva.badescu@umfiasi.ro; 8Department of Internal Medicine, “Grigore T. Popa” University of Medicine and Pharmacy, 700115 Iasi, Romania

**Keywords:** atherogenic dyslipidemia, metabolic syndrome, cholesterol, triglyceride, genetic risk factors, GWAS

## Abstract

Atherogenic dyslipidemia plays a critical role in the development of metabolic syndrome (MetS), being one of its major components, along with central obesity, insulin resistance, and hypertension. In recent years, the development of molecular genetics techniques and extended analysis at the genome or exome level has led to important progress in the identification of genetic factors (heritability) involved in lipid metabolism disorders associated with MetS. In this review, we have proposed to present the current knowledge related to the genetic etiology of atherogenic dyslipidemia, but also possible challenges for future studies. Data from the literature provided by candidate gene-based association studies or extended studies, such as genome-wide association studies (GWAS) and whole exome sequencing (WES,) have revealed that atherogenic dyslipidemia presents a marked genetic heterogeneity (monogenic or complex, multifactorial). Despite sustained efforts, many of the genetic factors still remain unidentified (missing heritability). In the future, the identification of new genes and the molecular mechanisms by which they intervene in lipid disorders will allow the development of innovative therapies that act on specific targets. In addition, the use of polygenic risk scores (PRS) or specific biomarkers to identify individuals at increased risk of atherogenic dyslipidemia and/or other components of MetS will allow effective preventive measures and personalized therapy.

## 1. Introduction

Atherogenic dyslipidemia is defined by increased levels of plasma triglycerides (TG > 150 mg/dL), small dense low-density lipoprotein cholesterol (LDL-C) > 130 mg/dL, total cholesterol (TC) > 200 mg/dL, apolipoprotein B (apoB) and free fatty acids (FFAs) associated with low HDL cholesterol level (HDL-C < 40 mg/dL) and is a major component of metabolic syndrome (MetS) [1]. Metabolic syndrome (MetS) encompasses a group of metabolic disorders that include dyslipidemia, central obesity, hyperglycemia/diabetes mellitus, and hypertension [2,3,4] (Figure 1).

The characteristic clinical manifestations of MetS are considered important risk factors for premature cardiovascular diseases (CVD). The prevalence of MetS is estimated between 13 and 36% among the European population [7].

MetS in Europe was 24.3% (23.9% in men vs. 24.6% in women, *p* < 0.001) in a study by Scuteri et al. [8]. The authors studied 34,821 subjects from 12 cohorts from 10 European countries and 1 of the American participants in the MARE (Metabolic Syndrome and Arteries Research) consortium with the aim of investigating differences in the distribution of “risky” clusters of MetS components according to geographic region and ethnicity. About 12% of subjects with MetS in Europe showed preexisting cluster triglyceride (T)-elevated BP (B)-waist circumference (W) (T-B-W group). This association was more frequent in patients from the UK (32.3%), Italy (19.6%), and Germany (18.5%), compared to Spain (2.6%), Sweden (1.2%), and the USA (2.5%). The G-B-W group was detected in 12.7% of patients with MetS, being more frequent in the population of Southern Europe compared to Northern Europe [8].

The prevalence of MetS (using The National Cholesterol Education Program (NCEP) Adult Treatment Group III (ATP III)) in the United States was 24%, and of the individual components of MetS was 30.0% for hypertriglyceridemia, 37.1% for low levels of HDL cholesterol, 34.0% for hypertension, 38.6% for increased waist circumference (WC), and 12.6% for hyperglycemia [9]. The estimate of the prevalence of MetS in the US National Health and Nutrition Examination Survey (NHANES) 2011–2018 revealed that it increased from 37.6% (95% confidence interval (CI): 34.0–41.4%) in 2011–2012 to 41.8% (95% CI: 38.1–45.7%) in 2017–2018. Among the MetS components, it was estimated that in the case of elevated plasma glucose values, the prevalence increased from 48.9% (95% CI: 45.7–52.5%) in 2011–2012 to 64.7% (95% CI: 61.4–67.9%) in 2017–2018 [10].

The etiopathogenesis of MetS is complex, heterogeneous, and not fully known, being the result of the interaction between genetic and environmental factors [11,12]. In many studies, CVD risk factors are classified into two categories: non-modifiable risk factors (age, sex, family history of CVD) and modifiable risk factors (sedentary lifestyle, smoking, hypercholesterolemia, diabetes mellitus (DM), systolic hypertension) [11,12].

Along with modifiable environmental factors (excess food and sedentary lifestyle), a major role in the etiology of MetS is played by genetic susceptibility (heritability). In recent years, many studies have focused on the study of genetic factors associated with the phenotypic manifestations of MetS. Another metabolic disorder specific to the MetS is insulin resistance. Insulin resistance is associated with increased fasting blood glucose and increases the sensitivity of visceral adipocytes to lipolytic hormones, which causes an increased flow of FFAs to the liver, stimulating hepatic triglyceride and ApoB synthesis [8,13]. Lipoprotein lipase (LPL) mediates the formation of LDL, a major element of dyslipidemia, in muscle tissue and adipose tissue [8,13]. The pathogenic mechanisms involved in the etiology of MetS are complex and most likely the result of the complex interaction between genetic and environmental factors at the level of different types of cellular structures. Over time, numerous candidate genes involved in the regulation of lipid metabolism have been discussed (e.g., polymorphism of the adiponectin, *PPARγ*, *LPL*, and *CETP* genes) [8].

The aim of our review is to provide a comprehensive analysis of data from the literature regarding the role of genetic factors involved in the etiology of atherogenic dyslipidemia associated with MetS. We also followed the interaction between these factors with other genes (epistasis) involved in the occurrence of other components of MetS or with environmental factors.

We analyzed the methods used in the study of the genetic factors involved in the etiology of atherogenic dyslipidemia and their results, emphasizing the advantages and disadvantages of each type of study. We also highlighted the perspectives that these data offer, both in terms of early identification of people at high risk for dyslipidemia (through the use of specific biomarkers or polygenic risk scores (PRS)) and the implementation of effective prevention measures, as well as for the development of innovative therapies that act on specific targets.

## 2. Strategies and Methods Used to Collect Data from the Literature

The data presented in the synthesis in this review were obtained through literature analysis (MEDLINE, PubMed, Google Scholar OMIM, MedGen databases) using the following keywords: atherogenic dyslipidemia, metabolic syndrome, atherosclerosis, genetic risk factors, heredity, monogenic dyslipidemia, polygenic dyslipidemia, candidate gene-based association studies (CGS), linkage-studies (LS), genetic linkage-analysis (LA), genome-wide association studies (GWAS), or whole exome sequencing (WES) (Table 1).

## 3. The Role of Atherogenic Dyslipidemia in the Metabolic Syndrome

Metabolic syndrome comprises an association of cardiovascular and metabolic risk factors, which include atherogenic dyslipidemia as a major element, along with increased blood pressure and plasma glucose [3,13] (Figure 2).

In addition, MetS is associated with a prothrombotic status, characterized by increases in both fibrinogen and plasminogen activator inhibitor-1 (increases in fibrinogen and plasminogen activator inhibitor-1) and proinflammatory, with increases in acute-phase reactants (C-reactive protein) [13].

Atherogenic dyslipidemia manifests predominantly as elevated levels of small low-density lipoprotein (LDL) particles and very-low-density lipoprotein (VLDL) remnants associated with reduced levels of high-density lipoprotein cholesterol (HDL-C) [93,94].

Atherogenic dyslipidemia is not usually characterized by elevated plasma levels of LDL cholesterol (LDL-C) because LDL particles are partially depleted of cholesterol. Plasma levels of VLDL remnants, intermediate-density lipoprotein (IDL), lipoprotein a Lp[a], and small LDL particles, which contain apolipoprotein B (ApoB) as the main protein, are increased. Furthermore, LDL precursors, especially remnant lipoproteins, are increased and appear to be as pro-atherogenic as LDL-C [13,93].

The prothrombotic and proinflammatory status associated with atherogenic dyslipidemia derives largely from the secretory activity of adipose tissue. Initially considered to be an inactive tissue, today, it is known that adipocytes release cytokines and other inflammatory markers, such as interleukin-6, tumor necrosis factor-alpha (TNF-R), and adiponectin. Adiponectin has the ability to reduce insulin resistance and may have antiatherogenic properties. Low adiponectin levels and, by implication, low atherogenic protection were associated with weight gain (particularly increased abdominal adiposity), whereas weight loss was associated with increased adiponectin levels and, presumably, increased atherogenic protection. In addition, plasma adiponectin levels vary inversely with fasting insulin levels and fasting plasma glucose (FPG) [13,93].

Abdominal adipose tissue shows an accelerated lipolytic activity (especially at the visceral level) with increased release of free fatty acids (FFAs), negatively influencing the action of insulin and the elimination of glucose at the tissue level [13].

The FFAs released from visceral adipose tissue reach the portal circulation and negatively influence the sensitivity of the liver tissue to insulin, the consequence being the increase in hepatic glucose synthesis. Subsequently, the increased plasma values of FFAs cause the accumulation of triglycerides in the muscle and liver tissue, thus reducing the action of insulin and increasing the production of lipoproteins containing apoB [13].

In this way, atherogenic dyslipidemia, together with other components of MetS (hyperglycemia), contribute to the atherogenesis process. Taking into account this aspect, patients with atherogenic dyslipidemia, in which other metabolic risk factors are also present, should be evaluated in terms of the risk for cardiovascular diseases (CVD) and their complications, with it being necessary to develop a plan for prevention and early treatment [3,11,12,94].

The National Cholesterol Education Program (NCEP) Adult Treatment Group III (ATP III) designated a group of metabolic (lipid and nonlipid) risk factors for cardiovascular disease (CVD). ATP III defined the MetS in the presence of more than 3 out of 5 clinical criteria, emphasizing the importance of insulin resistance and abdominal obesity in the etiology of MetS [2,95] (Figure 3).

The diagnosis of MetS was established in the presence of obesity and two of three of the following criteria: high blood pressure (BP), impaired glucose metabolism, and elevated non-HDL-C level (atherogenic dyslipidemia) [2,95].

The definition of MetS in simple clinical terms represented a diagnostic tool that would allow the identification of people who present a high risk of CVD in order to implement effective prophylactic measures through intervention on modifiable risk factors, but also early treatment to improve the prognosis [2,95].

Atherogenic dyslipidemia is frequently encountered in patients with type 2 diabetes mellitus (T2DM) and MetS. One of the most effective methods to reduce the risk of CVD is to modify the lipid profile. According to ATP III recommendations, since most atherogenic lipoproteins containing apoB are present in the LDL fraction, lowering LDL is the first priority. The most used drugs are statins; however, many studies have shown that despite intensive treatment with statins, the risk of acute coronary accidents remains high in many patients [96].

## 4. Heritability of Metabolic Syndrome and Its Components: Current Knowledge

The contribution of genetic factors (heritability) to the occurrence of MetS and its components has been proven by many family-based studies and twin studies [13]. In the study by Carmelli et al. [97], which included 2508 male twin pairs, the concordance for the three MetS components (DM, obesity, and hypertension) was 31.6% for monozygotic twins and 6.3% for dizygotic twins [97].

Lin et al. [95] showed, in a study that included 803 subjects from 89 Caribbean-Hispanic families, that the heritability of MetS (defined in accordance with the National Cholesterol Education Program Adult Treatment Panel III—NCEP/ATPIII criteria) was 24% (*p* = 0.009), and ranged from 16 to 60% for its five components [95]. The authors demonstrated moderate and significant heritability both for MEtS itself and for its individual components and independent factors that influence MetS [95].

Bellia et al. [98] showed, in the Linosa Study (LiS) group consisting of 293 Caucasian native subjects from 51 families (123 parents; 170 offspring), that the heritability of MetS was 27% (*p* = 0.0012), and among its individual components, heritability ranged from 10% for blood glucose to 54% for HDL-C. The MetS subtype associated with central obesity, hypertriglyceridemia, and Iow HDL had the highest heritability (31%; *p* < 0.001) [98].

The marked variability of MetS heritability in different studies could be partly attributed to the race and ethnicity of the individuals included in the study [13,98,99,100]. Starting from the evidence regarding heritability in the case of MetS, but also of its individual components, various studies were carried out that aimed to identify the determining genetic factors both in the case of MetS and its individual components (dyslipidemia, obesity, hypertension, and DM) [13,99,100].

Genetic factors act independently of environmental factors involved both in the etiology of atherogenic dyslipidemia and the other components of MetS, and the variable phenotype is the result of their permanent interaction. The study of monogenic mutations causing atherogenic dyslipidemia as well as polygenic risk factors (genetic polymorphisms and environmental factors) has been carried out by two types of studies: linkage analysis (LA) and association studies (CGS, GWAS, and WES) [101,102].

Linkage analysis (LA) studies investigate the association (co-segregation) between DNA polymorphisms and the onset of the disease and the hereditary inheritance of the disease in the studied family. LA studies are used to identify the involved gene locus and the pathogenic allelic variant, especially in the case of monogenic diseases and less so in the case of multifactorial (polygenic) diseases with complex etiology [102,103].

Association studies using the candidate-gene approach (CGA) represent an alternative method of studying polygenic diseases. CGA analyzes the association between genes and certain diseases, starting from the known pathogenic mechanisms of the disease in which these genes are supposed to intervene. The genes involved in lipid metabolism, obesity, DM, lipoproteins, and hypertension have been analyzed in MetS. Insufficient data on the highly complex pathogenic mechanisms of MetS and its components, however, limit this type of approach. The accelerated development of molecular genetics techniques in recent years has made extensive analyses possible, such as genome-wide association studies (GWASs) and whole exome sequencing (WES), which have provided new information related to the genetic component (heritability) in the case of multifactorial diseases (polygenic) [102,103].

Co-segregation of markers located near candidate genes for MetS and its components helps to identify the loci where those genes (called “positional candidate” genes) are located. The lack of uniform use of the criteria for defining MetS, as well as the multiple combinations between its components (which could be due to several loci specific to a certain association), represent the biggest disadvantage of using LA in the case of MetS. Furthermore, MetS is the consequence of the simultaneous action of multiple genes located on different chromosomes, masking the detection of discrete and unique linkage signals. The studies conducted so far have detected numerous loci involved in the etiology of MetS, located on chromosomes 1q23-31, 3q27, 17p12, chromosome 6 (D6S403, D6S264), and chromosome 7 (D7S479-D7S471) [104,105,106].

In a study that analyzed 2209 individuals with MetS, Kissebach et al. [104] showed that the 3q27 locus was strongly associated with six MetS phenotypic traits, including weight, body mass index (BMI), waist circumference (WC), hip circumference, insulin, and insulin-to-glucose ratio [104].

In the Insulin Resistance Atherosclerosis Family Study (IRAS FS), which included 216 Hispanic individuals with MetS, Langefeld et al. [105] provided evidence for the involvement of the 1q23-q31 locus [105].

In a study that examined patients from North America, Pollex et al. [107] identified several chromosomal regions (1p34.1, 1q41, 2p22.3, 7q31.3, 9p13.1, 9q21.1, 10p11.2, and 19q13.4) related to the occurrence of MetS [107].

A study that included 64 Chinese families revealed a significant association between MetS and its components in the case of some loci located on chromosomes 1, 2, and 16. Furthermore, a region located on chromosome 1q21-q25, known to be linked to T2DM in certain ethnicities, had a significant association with MetS [108].

Other studies that analyzed MetS components individually showed an association between the 5q locus and diastolic blood pressure (DBP); an association between loci 2q, 3q, 6q, 9q, 10q, and 17q and plasma levels of TG; while loci 12p, 12q, and 22q were associated with the plasma level of HDL-C [103,108].

In a study that included 250 German families, Hoffmann et al. [109] suggested that the 1p36.13 locus represents a susceptibility locus for T2DM and MetS [109].

Four other loci located on chromosomes 3p, 3q, 4q, and 14p were associated with T2DM, CVD, and MetS in the Diabetes Heart Study, which included 977 Caucasian subjects [103,110].

The results of these studies demonstrate that none of these loci can be definitively linked to MetS, there being differences at the population level, probably related to ethnicity and race, but also the lack of uniform criteria for defining MetS or false positive results. Taking into account these aspects, the results of LA studies exploring the genetic causes of complex diseases, such as MetS, must be interpreted with caution [103,109,110].

## 5. Monogenic Etiology of Atherogenic Dyslipidemia

Low-density lipoprotein cholesterol (LDL-C) molecules play a major role in the initiation and progression of atherosclerotic plaques, increasing the risk of acute vascular events (AVEs). Thus, mutations of genes involved in lipid metabolism, associated with the increase of plasma LDL-C levels, make a major contribution to the occurrence of atherosclerotic cardiovascular disease (ASCVD) and its complications [15].

### 5.1. Genetic Mutations Associated with the Increase in the Plasma LDL-Cholesterol Level

#### 5.1.1. Genes That Determine Familial Hypercholesterolemia Type 1 (FH)

Familial hypercholesterolemia (FH) (OMIM, 606945) [14] has been associated with mutations in genes encoding the LDL receptor (LDLR), the apolipoprotein B-100 (ApoB-100, a ligand of the LDL-C receptor), and proprotein convertase subtilisin kexin type 9 (proprotein convertase, subtilisin/kexin-type 9, PCSK9). FH is an inherited metabolic disease associated with elevated plasma LDL-C levels and premature CVD [14,15].

Low-Density Lipoprotein Receptor (*LDLR*) gene

Mutations of the *LDLR* gene (located on chromosome 19p13.2) cause about 85–90% of FH cases. Most cases of FH are caused by heterozygous mutations (HeFH) transmitted in an autosomal dominant or codominant manner, with an estimated prevalence of 1 in 250 subjects worldwide [14,16]. The disease is characterized by genetic heterogeneity. Currently, 3839 mutations have been described, distributed throughout the entire *LDLR* gene [14], which determine the synthesis of an abnormal or non-functional protein (LDLR). The LDLR protein plays an important role in maintaining cholesterol homeostasis, being involved in both absorption and degradation of LDL-C. Homozygous mutations in the *LDLR* gene cause a very severe form of FH, an autosomal recessive disease (HoFH), and are associated with plasma LDL-C levels approximately 6–10 times higher (600–1200 mg/dL) detected from birth; they are also associated with increased risks for premature ASCVD and death [17,18]. Homozygous FH (HoFH) is characterized by a much lower prevalence, around one case per 160,000–300,000 individuals [16]. In the absence of treatment, patients with HoFH present their first episode of ASCVD in childhood or adolescence, while patients with HeFH usually experience their first ASCVD event in the third or fourth decade of life [16,19]. It is proven that elevated plasma LDL-C levels throughout life are the main determinant of increased risk of ASCVD in patients with HF [19]. The coronary vascular bed is most frequently affected in patients with FH, but sometimes cerebrovascular or peripheral artery damage is also detected [16,19].

b.*APOB* gene

Apolipoproteins ApoB-100 are the main protein components of apolipoproteins synthesized in the liver (chylomicrons (CM), very-low-density lipoproteins (VLDL), and low-density lipoproteins (LDL)) and ApoB-48 (synthesized exclusively in the intestine). They are encoded by the *APOB* gene, located on chromosome 2p24.1 [14]. ApoB-100 binds to heparin and different proteoglycans from the arterial walls and has an LDLR binding domain, intervening in the regulation of plasma levels of cholesterol by eliminating LDL-C from the body [20].

Familial hypercholesterolemia type 2 (FHCL2) (OMIM 14401014), also called familial defective apolipoprotein B-100 (FDB), is an autosomal dominant disorder manifested by increased plasma lipids level and early onset of atherosclerosis [14]. The decreased affinity for LDLR present in FHCL2 is caused by two *APOB* gene allelic variants: C10580G (p. Arg3527Gln) [21] and C10800T (p. Arg3531Cys) [22].

Thomas et al. [111] identified another mutation (p.Arg50Trp) in exon 3 of the *APOB* gene in a study that combined genetic linkage analysis (LA) with WES in members of a family with FH (familial hypercholesterolemia type 1, AD) [111].

Other *APOB* mutations cause abetalipoproteinemia (ABL) (OMIM, 200100) and hypobetalipoproteinemia (FHBL) (OMIM, 615558), two rare, autosomal recessive disorders characterized by hypocholesterolemia and malabsorption of fat-soluble vitamins, causing neuropathy, coagulopathy, and retinal degeneration [14].

Kristiansson et al. [24] first reported the association between *APOB* allelic variants rs673548 and rs6728178 and increased risk of MetS [24].

Later, two GWASs analyzed in the Caucasian population identified two other allelic variants involved in MetS: rs673548, identified by Lind et al. [25] and rs1367117, identified by Guindo-Martínez et al. [26]. Jang et al. [27] identified two new *APOB* allelic variants (rs4665709 and rs144467873) in MetS patients from Taiwan [27].

c.*PCSK9* gene

The identification of the *PCSK9* gene, which plays an essential role in FH and cholesterol metabolism, created the premise for the development of new targeted gene therapies [14].

A gain-of-function (GOF) mutation in the *PCSK9* gene (p.(Ser127Arg)) was identified in two large French families with multiple consanguineous hypercholesterolemia/hyper LDL-C, tendinous xanthomas, acute myocardial infarction (MI), and stroke, which most likely had a common ancestor. Another French patient (died of MI) in whom extremely elevated plasma LDL-C levels were detected had a mutation in exon 4 of *PCSK9* p. (Phe216Leu) [28]. The pathogenic *PCSK9* p.(Asp374Tyr) allelic variant was subsequently identified in 18 patients from 31 families in Utah, the mutation being subsequently detected in other Norwegian and English patients and associated with a severe phenotype [28].

The loss-of-function mutations (LOF) in the *PCSK9* p.(Tyr142*) and p.(Cys679*) were identified in 2.6% of African–American patients analyzed in the study by Cohen et al. [29] and were associated with a 28% reduction in mean LDL-C and an 88% reduction in coronary heart disease (CHD) risk [29]. The rare allelic variant p.(Arg46Leu) was identified in Caucasian subjects analyzed in the ARIC study and was associated with a 15% reduction in plasma LDL-C levels and a 47% decrease in CHD risk [28,29]. Subsequent studies on Caucasians identified other allelic variants p.(Gly106Arg), p.(Arg237Trp), and p.(Arg46Leu), which have hypocholesterolemic effects [28]. However, their general effect on the blood levels of TC and LDL-C was lower compared to the effect of the genetic variants identified in African–American subjects [28].

*PCSK9* gene polymorphism is associated with both plasma lipid levels and response to statin therapy. The Atherosclerosis Risk in Communities (ARIC) and the Dallas Heart Study showed that some LOFs in the PCSK9 gene have a cardioprotective effect [29].

Chuan et al. [30] showed that the *PCSK9* rs562556 (c.1420G>A, I474V) polymorphism (located in exon 9) was associated with low plasma levels of TC and LDL-C [30]. Thus, these findings were the basis for the introduction of new therapies, such as PCSK9 inhibitors, aimed at reducing plasma cholesterol levels [30].

d.LDLRAP1 gene

LOFs in the *LDLRAP1* (LDL receptor adapter protein 1) gene (located on chromosome 1p34-1p35) are very rare and lead to the synthesis of a non-functional or truncated LDLRAP1 protein. LDLRAP1 protein is required for LDLR internalization in hepatocytes [14]. Homozygous pathogenic mutations in the *LDLRAP1* gene or the compound heterozygous genotype are associated with the occurrence of a severe form of autosomal recessive hypercholesterolemia (ARH/FHCL4) (OMIM, 603813) [14]. The particularly severe phenotypic manifestations of ARH are similar to those of HoFH (homozygous familial hypercholesterolemia) and are associated with an increased risk of ASCVD with rapid lethal evolution despite treatment. A treatment that seems efficient in the case of ARH consists of a combination of lipid-lowering drugs that reduce plasma LDL-C levels and Lomitapide [14].

To date, 50 cases of ARH are described in the literature, the patients coming from the Mediterranean basin or the Middle East [14]. Arca et al. [31] analyzed 28 people from 17 unrelated families from Sardinia in which they identified two mutations in the *LDLRAP1* gene: a frameshift mutation C432insA (p.FS170stop) in exon 4 (ARH1) and a nonsense mutation C65G->A (p.Trp22ter) in exon 1 (ARH2) [31]. Although initially, the ARH1 homozygous genotype was identified only in the case of four patients from mainland Italy [31], subsequent studies allowed the identification of new pathogenic mutations. Nikasa et al. [32] identified a new pathogenic homozygous mutation, c.649G>T, p.Glu217Ter, in exon 7 of the *LDLRAP1* gene, causing severe ARH in a 20-year-old patient of Iranian origin [32].

Feng et al. [33] identified a novel homozygous *LDLRAP1* variant (c.649G>T, p.Glu217Ter) in exon 7, associated with a severe form of ARH1 in a Chinese girl. The 13-year-old patient presented typical symptoms of nephrotic syndrome with an abnormally high plasma LDL-C level [33]. Renal biopsy identified the presence of a membranous nephropathy (MN). Renal biopsy suggested that the nephrotic syndrome was induced by MN, with no evidence of secondary MN found. The medical treatment consisted of aggressive lipid-lowering therapy associated with angiotensin receptor blockers leading to remission of proteinuria and stabilization of the clinical condition during 2 years of follow-up [33].

#### 5.1.2. Genetic Mutations Associated with Low Plasma HDL-Cholesterol Level

Many prospective studies revealed that low plasma high-density lipoprotein cholesterol (HDL-C) level (HDL-C < 35–40 mg/dL according to current guidelines) correlates negatively with the incidence of CHD [112,113]. The prevalence of low HDL-C levels in adult patients in the United States was 19%, with variations by gender (35.4% for men and 11.8% for women) and ethnicity [114].

There is evidence that HDL-C has a protective role against CHD, given its role in reverse cholesterol transport and other atheroprotective effects that are associated with this class of lipoproteins. Although some clinical studies suggest that increasing HDL-C level would have a beneficial effect in reducing the risk of atherosclerosis, a recent study showed that in the case of the cholesteryl ester transfer protein (CETP) inhibitor torcetrapib, no positive effect related to coronary atherosclerosis was identified, despite the fact that the treatment showed a substantial increase in HDL-C and a decrease in LDL-C levels [112]. In a large multiethnic population study, individuals with low HDL-C levels had a significantly increased risk for MetS. The inverse association between plasma HDL-C level and MetS incidence was independent of traditional risk factors for MetS, including visceral adiposity, insulin resistance, and inflammatory syndrome [112].

*APOA1* gene

Apolipoprotein AI (ApoA-I) is a cofactor of the enzyme lecithin-cholesterol acyltransferase (LCAT), which determines the esterification of cholesterol, thus favoring its removal from plasma and tissues. ApoA-I is encoded by the *APOA1* gene and is located on chromosome 11q23.3 [14].

Some allelic variants of the *APOA*1 gene (homozygous, heterozygous, or compound heterozygous genotype) cause familial hypoalphalipoproteinemia AD (hypoalphalipoproteinemia, Primary, 2, OMIM 618463), which includes two autosomal recessive diseases: the combined deficiency of Apo-I and apoC-III (ApoA-I and apoC-III deficiency, combined) [14,34].

The *APOA1* gene polymorphism is associated with low plasma HDL-C levels and an increased risk of early-onset ASCVD. Homozygous mutations in the *APOA1* gene lead to the complete absence of ApoA-I associated with low plasma HDL-C levels < 5 mg/dL and normal plasma levels of LDL-C and TG. Heterozygous missense mutations in the *APOA1* gene affect the structure and, sometimes, the function of the ApoA-I protein, causing a decrease in the plasma levels of ApoA-I and HDL-C [34].

In a study conducted on a group of 67 Japanese children with low plasma HDL-C levels, Yamakawa-Kobayashi et al. [35] analyzed the *APOA1* gene polymorphism [35]. They identified four different mutations (three frameshifts mutations and one splice site mutation) in four of the children, associated with plasma ApoA-I levels reduced by approximately 50% of the normal value. The authors estimated that in the Japanese population, the frequency of hypoalphalipoproteinemia caused by *APOA1* mutations is 0.3% in the general population and 6% in individuals with low plasma HDL-C levels [35].

Haase et al. [35] showed that certain rare *APOA1* allelic variants (e.g., the A164S variant) determine a predisposition to amyloidosis associated with low plasma levels of ApoA-I and HDL-C, with carriers having an increased risk of CVD and CAD, in a study performed on the Danish population. Compared to the control group, life expectancy was decreased by more than 10 years (*p* < 0.0001) [36].

It has been proven that the rare cysteine variants of ApoA-I, ApoA-IMilano (apoA-I(Milano)), and ApoA-I(Paris) have a cardioprotective effect, although they cause a decrease in plasma HDL-C levels and an increase in plasma TG levels. In addition, they can decrease HDL-C levels even in the absence of CVD [37].

Kim et al. [72] analyzed 835 Korean subjects (320 MetS patients and 515 healthy controls), looking at the association of two *APOA1* polymorphisms (XmnI and MspI/rs670) with their susceptibility to MetS. The authors showed that the two allelic variants of *APOA1* were associated with a lower risk of MetS. The X1-A and X2-G haplotypes for the *APOA1* XmnI and MspI polymorphisms were also protective against the risk of MetS [72].

The two allelic variants (XmnI and MspI) had a significant association with body mass index (BMI) (*p* = 0.037) and DBP (diastolic blood pressure) (*p* = 0.047). From these results, the authors concluded that the analyzed *APOA1* polymorphisms and haplotypes may have a protective role against MetS susceptibility in the Korean population [72].

Bora et al. [38] showed that the *APOA1* G-75A and C+83T polymorphisms were associated with CVD risk in the Indian population of Assam. Thus, the A allele at G-75A was pro-atherogenic, and the C allele at the C+83T locus was anti-atherogenic. The C+83T locus was associated with plasma TG and VLDL-C levels, and the G-75A locus was associated with LDL-C levels. The two loci were not associated with plasma HDL-C levels. No significant association of the two loci with obesity, TC, or BP was found, although there was a significant association of the two loci with several pro-atherogenic factors [38].

In a study that included 1267 elderly people (aged over 65 years) of Chinese origin, Nie et al. [39] found a significant correlation between a high TG/HDL-C ratio and an increased risk of MetS in the analyzed population [39].

b.*ABC1* gene

The ATP-binding cassette transporter 1 (ABCA1) is a membrane protein that mediates the extracellular transport of cholesterol and phospholipids, having a protective role against atherosclerosis. ABCA1 is encoded by the *ABC*1 gene, located on chromosome 9q31 [14]. *ABCA1* gene mutations cause a severe deficiency of the encoded protein and could contribute to the atherogenesis process associated with metabolic disorders and common inflammatory diseases [14]. Homozygous or compound heterozygous mutations in the *ABCA1* gene cause Tangier disease (TGD) (OMIM, 205400). TGD is a rare autosomal recessive disease characterized by extremely low plasma HDL-C levels (HDL-C < 5 mg/dL and ApoA-I ≤ 10 mg/dL) and an increased risk of early CAD [14]. TGD is manifested by peripheral neuropathy, hepatosplenomegaly, orange-colored pharyngeal tonsils, corneal opacities, and lymphadenopathy [41].

In the literature, 331 mutations of the *ABCA1* gene are reported. A heterozygous carrier of the LOF mutation in the *ABCA1* gene can have variable plasma HDL-C and ApoA-I levels, sometimes being reduced by up to 50% compared to normal values [41].

The *ABCA1* R219K polymorphism (G1051A, rs2230806) K allele was associated with increased plasma HDL-C levels and may be associated with a reduced risk of ASCVD in Asians (including Japanese) and Caucasians [42].

c.*LCAT* gene

The enzyme lecithin cholesterol acyl transferase (CAT) facilitates the removal of cholesterol from plasma and tissues and is encoded by the *LCAT* gene (located on chromosome 16q22.1) [14].

LCAT intervenes in the maturation of HDL-C through a process of esterification of free cholesterol with acyl groups derived from lecithin [44,115]. There is the hypothesis that the LCAT enzyme has two different plasmatic activities: the alpha-LCAT activity is specific for HDL, and the beta-LCAT activity for VLDL and LDL [115].

LCAT enzyme deficiency, caused by homozygous or compound heterozygous mutations in the *LCAT* gene, causes the appearance of two autosomal recessive diseases: Norum disease (OMIM 245900) and fish-eye disease (FED/Partial LCAT deficiency) (OMIM 136120) [14]. Norum disease is a rare disease manifested by atherosclerosis, proteinuria, renal failure, corneal opacities, and hemolytic anemia. In Norum’s disease, both enzyme activities (alpha and beta) are lost, causing very low HDL-C levels, decreased LDL-C, and increased TG levels [45].

In FED, the alpha-LCAT activity is lost, while the beta activity is preserved, allowing the esterification of cholesterol into VLDL and LDL-C but not into HDL-C [115].

There are few follow-up longitudinal studies in which a heterogeneity of the molecular deficits associated with *LCAT* has been observed. Although LOFs in the *LCAT* invariably lead to a marked decrease in plasma HDL-C levels, the role of *LCAT* in atherogenesis remains controversial [45,115].

Currently, about 138 mutations are described, mostly in exons 1 and 4 of the *LCAT* gene, without being able to make a correlation between genotype and phenotype or with ethnicity [44]. Many of the population studies did not detect a correlation between low plasma LCAT levels and an increased risk for CVD, suggesting that reduced LCAT activity may not be a risk factor nor a therapeutic target.

More recent studies of heterozygous carriers of *LCAT* mutations tend to suggest that there is a genotype–phenotype correlation, with atherogenicity being correlated with the type of *LCAT* mutation, which could explain the conflicting results in some studies. Although the important effects of *LCAT* in HDL-C and TG metabolism are known, the role of *LCAT* in metabolic disorders, including obesity and DM, has not been sufficiently studied [14,44].

Recent studies in LCAT-deficient mouse models suggested that the absence of LCAT could have a protective effect against insulin resistance, diabetes, and obesity.

Based on these observations, future research could investigate the role of LCAT in the occurrence of metabolic disorders causing MetS [44].

#### 5.1.3. Genetic Mutations Associated with Hypertriglyceridemia

It is known that excess intra-abdominal adipose tissue is frequently associated with insulin resistance and increased atherogenic risk, specific manifestations of MetS [116]. Many studies have shown that excess accumulation of visceral adipose tissue is associated with increased plasma TG levels, decreased plasma HDL-C levels, and an increased proportion of small and dense LDL particles despite normal LDL-C. LDL-C and HDL-C are mainly involved in cholesterol transport, while chylomicrons (CM) and VLDL particles mainly transport TG. Environmental factors (e.g., diet, sedentary lifestyle, smoking) can also influence plasma TG levels [75,116,117].

There is evidence attesting to the fact that excess visceral adipose tissue present in obese patients is associated with resistance to insulin action and disturbances in plasma glucose homeostasis [116]. Abdominal obesity was also associated with the occurrence of hypertension and predisposition to inflammation and thrombosis. Furthermore, people with normal weight can have an excess of visceral adipose tissue and can manifest metabolic disorders specific to MetS [116]. Over time, many studies have demonstrated that elevated plasma TG levels are a strong independent risk factor for CVD [75,116].

*LPL* gene

Lipoprotein lipase (LPL) is an enzymatic protein that converts VLDL to LDL-C and is encoded by the *LPL* gene (located on chromosome 8p21.3) [14,46].

LPL is a water-soluble enzyme that hydrolyzes TG in lipoproteins (chylomicrons and VLDL). LPL requires apoC-II as a cofactor and is involved in stimulating the cellular absorption of chylomicron remnants, FFAs, and cholesterol-rich lipoproteins [14]. Polymorphisms of the *LPL* gene (D9N and N291S) have been shown to cause an increase in TC and TG and a decrease in HDL-C level, determining a phenotype specific to hypercholesterolemia [14,46]. Many GWASs have reported the association between SNPs in the *LPL* gene (rs12678919, rs328, rs10503669, rs17411031, rs10096633, rs17482753, rs2083637) and hypercholesterolemia, especially related to HDL-C level [47].

Homozygous or compound heterozygous mutations in the *LPL* gene cause lipoprotein lipase deficiency (LPL deficiency or type I hyperlipoproteinemia, or familial chylomicronemia syndrome, OMIM 238600), a rare autosomal recessive disorder [14]. Frequently, the disease manifests itself in childhood through episodes of acute abdominal pain, recurrent pancreatitis, hepatosplenomegaly, eruptive xanthomas, and early-onset atherosclerotic CAD, associated with severe hypertriglyceridemia, lactescent serum, and decreased plasma levels of LDL-C and HDL-C. A diet with restricted total lipids (≤20 g/day) usually leads to the remission of symptoms [14]. Heterozygous individuals may have mild hyperlipidemia and reduced postheparin plasma lipolytic activity (PHLA) without associated early atherosclerosis [14].

Common *LPL* gene mutations (Asp9Asn, Asn291Ser, Trp86Arg, Gly188Glu, Pro207Leu, Asp250Asn) associated with type I hyperlipoproteinemia have been detected in approximately 20% of patients with hypertriglyceridemia [48]. Taking into account this aspect, it is recommended that these mutations be tested, especially in patients with a high risk of premature atherosclerosis. Studies have shown that a relatively large number of individuals with low favorable lipids were carriers of common mutations (Ser447X) or silent mutations (Thr361) in the *LPL* gene [48].

Wittekoek et al. [49] showed that a common *LPL* variant (N291S) significantly influences the biochemical phenotype and risk for CVD in patients with FH [49].

Familial chylomicronemia syndrome (FCS) manifests at young ages and has a prevalence of 1:100,000 to 1:1,000,000 individuals. FCS is characterized by severe hypertriglyceridemia (>10 mmol/L) associated with elevated fasting chylomicron levels. Chylomicrons transport TGs to adipose tissue, skeletal and cardiac muscle, where they are broken down by the enzyme LPL [46]. Chylomicrons can block pancreatic capillaries and, together with severe hypertriglyceridemia, increase the risk of pancreatitis. Other clinical manifestations include eruptive xanthomas, hepatosplenomegaly, neurological symptoms, and lipemia retinalis [46]. FCS is caused by homozygous or compound heterozygous mutations in the *LPL* gene detected in over 80% of patients. In other cases, there are detected mutations of the *APOA5, APOC2, GPIHBP1,* or *LMF1* genes that affect the function of the LPL enzyme, causing high levels of chylomicrons in the plasma associated with low plasma levels of lipoproteins. The *APOA5* and *APOC2* genes have a role in the activation of the *LPL* gene; the *LMF1* gene intervenes in the transport of activated LPL, and *GPIHBP1* anchors LPL to the surface of capillary endothelial cells [46].

b.*APOC2* gene

The *APOC2* gene (located on chromosome 19q13.32) encodes apolipoprotein C-II (ApoC-II), a cofactor required for the activation of LPL, the enzyme that hydrolyzes plasma triglycerides and transfers FFAs to tissues [14]. Homozygous mutations in the *APOC2* gene are associated with hyperlipoproteinemia, type Ib (OMIM, 608083), an autosomal recessive disease characterized by extremely high serum concentrations of TG (up to 30,000 mg/dL) and chylomicrons (CM), clinically manifested by recurrent pancreatitis eruptive xanthomas and hepatomegaly. In addition to the hereditary form, an acquired, reversible form of ApoC-II deficiency is described, frequently caused by chemotherapy and which is not associated with an increased risk of CHD [46].

c.*ABCG5* and *ABCG8* genes

Mutations of the *ABCG5* (encodes sterolin-1) and *ABCG8* (encodes sterolin-2) genes located on chromosome 2p21 cause sitosterolemia (STSL) [14]. The increased intestinal absorption of both cholesterol and sterols from plants and crustaceans, and the reduced capacity for biliary excretion of sterols, determine high levels of plant sterols in plasma (>30 times the normal value) associated with the appearance of tendinous xanthomas and premature ASCVD [14,50].

Most STSL patients have homozygous or compound heterozygous mutations of the two genes involved. The prevalence of heterozygous mutations and the phenotypic peculiarities associated with them are not exactly known [51]. In the case of Caucasian patients, *ABCG8* mutations were frequently detected, while in Asian patients, *ABCG5* mutations were frequently detected. In a study of the Chinese population, Wang et al. [50] reported the presence of *ABCG8* mutations in three of the eight families studied, suggesting that *ABCG8* mutations are not exclusively present in Caucasians [50].

d.Glycerol-3-phosphate dehydrogenase-1 (*GPD1*) gene

Mutations in the glycerol-3-phosphate dehydrogenase 1 (*GPD1*) gene (located on chromosome 12q13.12) are the cause of transient infantile hypertriglyceridemia (hypertriglyceridemia, transient infantile, HTGTI) [14]. HTHT1 is an autosomal recessive genetic disorder characterized by abnormal lipid synthesis and excessive secretion of TG in the liver. At birth, moderate/severe hypertriglyceridemia is frequently detected, which is associated in the neonatal period with hepatomegaly, persistent hepatic steatosis, and the beginning of hepatic fibrosis. However, in childhood and adolescence, plasma TG levels tend to normalize [46].

e.AGPAT2, BSCL2, CAV1, and CAVIN1 genes

Monogenic congenital lipodystrophy syndromes (CGLS) represent a heterogeneous group of diseases characterized by a total or partial loss of body fat (lipoatrophy) caused by the inability to store fat in adipose tissue and frequently associate severe hypertriglyceridemia. The most common types are isolated congenital lipodystrophy (CGL) and familial partial lipodystrophy (FPLD), caused by mutations of genes involved in lipid and adipocyte metabolism, transmitted in a dominant or recessive manner [14,46].

The most frequently involved genes are *AGPAT2* (located on chromosome 9q34), encoding a key enzyme in triglyceride biosynthesis (1-acyl-glycerol-3-phosphate-O-acyltransferase-2) and *BSCL2* (located on chromosome 11q13), encoding the endoplasmic reticulum seipin protein. Some cases of CGLS are due to homozygous pathogenic mutations in the *CAV1* (located on chromosome 7q31.2) and *CAVIN1* (located on chromosome 17q21.2) genes. The proteins encoded by the two genes (caveolin-1 and cavin-1) are major components of the plasma membrane called caveolae. Homozygous pathogenic mutations in the *PPARG* (3p25.2) or *LMNA* (1q22) genes cause a form of generalized lipodystrophy, while heterozygous carriers manifest a form of familial partial lipodystrophy (FPLD) [52].

CGL is characterized by the excess deposition of TG in non-adipose tissues, where they induce a lipotoxic activity. Patients have insulin resistance, DM, and hepatic steatosis. Monogenic lipodystrophies are rare diseases, with prevalence varying from 1:20,000 to less than 1:10,000,000, but there are also forms acquired after drug treatment or autoimmune reactions. It is possible that individuals who develop a form of acquired lipodystrophy also have a genetic predisposition, but this has not yet been proven [14,46].

f.The *APOE* gene

Familial dysbetalipoproteinemia (FDBL) is most frequently determined by homozygous recessive mutations of the E2 allele of the apolipoprotein E (*APOE*) gene (located on chromosome 19q13.32). In ~10% of FDBL cases, dominant allelic variants E3 or E4 are detected [14,46].

Studies have revealed that only 5–10% of individuals homozygous for the *APOE2* allelic variant develop FDBL, the intervention of some favorable secondary factors, such as DM or high alcohol consumption, being discussed. The intervention of several genes (polygeny) is also discussed, the occurrence of FDBL being the consequence of the cumulative effect of multiple genetic risk variants that are common in the general population. The prevalence of FDBL is up to 1–2% in the general population, but often the disease remains undiagnosed [14,46].

ApoE mediates the binding of lipoprotein remnants (RPL) (e.g., remnants of chylomicrons and VLDL) or lipid complexes from the plasma and interstitial fluids to specific receptors on the cell surface, a process that leads to their elimination from the circulation. Due to deficient ApoE, RLP clearance is reduced and RLP plasma levels are increased, being associated with a strong atherogenic risk associated with premature CVD. Patients with FDBL frequently present palmar xanthomas and elevated plasma TG and TC levels (7 to 10 mmol/L) associated with decreased plasma HDL levels and a VLDL/triglyceride ratio greater than 0.3 [14,46].

#### 5.1.4. Familial Combined Hyperlipidemia and Familial Hypertriglyceridemia

The *APOA1/C3/A4/A5* Gene Cluster and Lipid Metabolism Disorders

Familial Combined Hyperlipidemia (FCHL)

Combined familial hyperlipidemia is considered the most common and less characterized atherogenic dyslipidemia, associated with a cardiovascular risk between three and ten times higher than that of the general population [53].

FCHL is the most common form of primary dyslipidemia, affecting 1–2% of the Western population and 14–20% of patients with premature CVD. FCHL manifestations are heterogeneous, being able to manifest in the form of mixed hyperlipidemia, isolated hypercholesterolemia, hypertriglyceridemia, or in combination with increased levels of ApoB [14]. Currently, FCH is defined as a common metabolic disorder characterized by: 1. increased plasma cholesterol and/or TG levels in at least two members of the same family; 2. variable expressivity of lipid phenotype in members of the same family; 3. increased risk of premature CHD [14,55].

Linkage analysis (LA) studies and GWASs have suggested that the etiology of FCHL is complex and multifactorial, although initially, FCHL was considered a monogenic autosomal dominant disease with incomplete penetrance. The characteristic FCHL phenotype is determined by the interaction between genetic factors (polygeny) and environmental factors. The etiology of FCHL is not fully understood, but GWASs have indicated three possibly involved loci located on chromosomes 1q21-23, 11p14.1-q12.1, and 16q22-24.1 [54,55].

Several studies have revealed that the *APOA1/C3/A4/A5* haplotype (located on chromosome 11p14.1-q12.1) plays an important role in lipoprotein metabolism [56]. Liu et al. [57] showed that certain variants of the *APOA1/C3/A4/A5* haplotype can be predictive markers of response to fenofibrate therapy, but these results need to be confirmed later by other studies [57].

The *APOA5* c.553G>T (rs2075291) polymorphism was associated with increased plasma TG levels and an increased risk for CAD [58]. The *APOA5* c.56G>G and *APOC3* c.386G>G allelic variants are associated with FCHL [59,60].

Patients with FCHL have an increased risk of CVD frequently associated with steatohepatitis, non-alcoholic fatty liver disease, T2DM, and MetS. Hopkins et al. [61] identified MetS in 65% of FCHL patients compared with 19% of control subjects [61].

b.Familial Hypertriglyceridemia (FHTG)

Familial hypertriglyceridemia (FHTG, OMIM 145750) is a rare, hereditary, primary dyslipidemia characterized by moderately elevated serum TG (>400 mg/dL), usually in the absence of significantly increased plasma cholesterol levels [14].

FHTG is a monogenic autosomal dominant disorder with a prevalence of 5–10% in the general population and is usually diagnosed in adulthood. The characteristic phenotypic manifestations of FHTG are obesity and decreased glucose tolerance. The main physiopathological mechanism of FHTG is the slow catabolism of VLDL rich in TG, secreted in the liver tissue. GWAs have suggested possible loci associated with FHTG located on chromosomes 15q11.2-q13.1 and 8q11-q13 [54].

Rare allelic variants of *APOA5* (associated with increased plasma TG) and *LDLR* (associated with increased plasma LDL-C levels) genes increase the risk of CAD and MI [62].

In a study that included Chinese patients with hypertriglyceridemia, Kao et al. [63] identified the association between the *APOA5* G553T allelic variant (causing the substitution of cysteine with glycine-185) and elevated plasma TG levels [63].

The prevalence of FHTG in families with premature CAD was analyzed in two independent studies. Genest et al. [64] identified the presence of hypertriglyceridemia in 1% of families with CAD diagnosed before the age of 60, while 14.7% of cases presented hypoalphalipoproteinemia [64]. Hopkins et al. [61] found the presence of FHTG in 20.5% of families in which there was at least one case of CAD; approximately 71% of patients with FHTG had MetS compared to 19% in the control group [61].

#### 5.1.5. Atherosclerosis Susceptibility/Atherogenic Lipoprotein Phenotype

Atherosclerosis susceptibility (ATHS), also called atherogenic lipoprotein phenotype (ALP) (OMIM 108725) [14], is an autosomal dominant monogenic disease caused by the mutation of the *ATHS* gene (located on chromosome 19p13.3-p13.2).

ATHS/ALP is associated with an increased risk of ASCVD and MI in the presence of increased plasma LDL and triglyceride-rich lipoprotein levels and reduced plasma HDL levels [14].

Variant A of ALP is characterized by the presence of large LDL particles, while in variant B, which is associated with an increased risk for CAD, small and dense LDL particles are detected [14].

Austin et al. [65] concluded that the B phenotype may be an independent risk factor for CAD and MI [65].

Nishina et al. [66] suggested that the *ATHS* gene that causes ATHS/ALP might be the same as the *LDLR* gene (located on chromosome 19p13.3-p13.2) or located near the *LDLR* locus [66].

Rotter et al. [67] highlighted the linkage between the *ATHS* gene and the *LDLR* locus, suggesting that ALP phenotypic manifestations are determined by a gene other than *LDLR* [67].

There is evidence that the *CETP* genes (located on chromosome 16q13) encoding cholesteryl ester transfer protein-CETP and the *SOD2* gene located on chromosome 6q25 (encoding superoxide dismutase-1—SOD2) could be linked to the *APOA1/APOC3/APOA4* haplotype. Although some studies have suggested that there are genetic factors common to those of FCH, only the B phenotype of ALP is associated with early CAD [68].

## 6. Polygenic Etiology of Atherogenic Dyslipidemia: Gene Polymorphisms Associated with Metabolic Syndrome in the GWASs Era

Linkage analysis (LA) studies allow the identification of monogenic diseases with Mendelian patterns of inheritance caused by large-effect allelic variants with low frequency. In contrast, extensive analyses such as GWAS can identify genes associated with diseases with complex, multifactorial (polygenic) etiology caused by interactions between genetic and environmental factors [99,102].

GWASs examine the co-segregation of polymorphic genetic markers (single nucleotide polymorphism—SNP) distributed throughout the genome in families affected by MetS or its components (atherogenic dyslipidemia, obesity, hypertension, and insulin resistance).

A rare allelic variant present in 1% of the population is considered a polymorphism. It is estimated that approximately 3,000,000 SNPs (one SNP in every 1000 base pairs) are present in the entire genome (3 billion base pairs) [99,102,103].

The progress made with the development of molecular technology, including GWAS, allowed the elucidation of the molecular substrate and physiopathological mechanisms in many of the complex, multifactorial diseases. Thus, GWASs allowed the identification of numerous genetic factors associated with MetS and its components (dyslipidemia, obesity, DM, and hypertension) [99,103].

Until 2014, only four GWASs using MetS or components of MetS were published, but their number subsequently increased in the following years.

Zabaneh et al. [118] carried out a GWAS in two stages, in which they aimed to identify the genetic factors involved in the occurrence of MetS and the phenotypes associated with MetS in a group of Indian male patients. Initially, approximately 317,000 SNPs were genotyped in 2700 individuals, of which 1500 SNPs were selected to be genotyped in another 2300 individuals [118].

The selection for the inclusion of patients in stage 1 was based on four component traits of MetS: HDL-C, plasma glucose, T2DM, abdominal obesity measured by waist-to-hip ratio, and DBP [118]. Common SNPs were analyzed for association with MetS and the four individual components of MetS. Four SNPs were identified (rs376426, rs9989419, rs4523270, rs496300) reaching a significance level of *p* < 5 × 10^(−7)^ and with a posterior probability of association (PPA) > 0.8 at the level of *CETP* and *LPL* genes associated with HDL-C. These associations were previously reported in the European population and in Asian Indians [118].

Other additional loci were identified (they had a significant association with *p* < 10^(−6)^ and posterior probability (PPA) > 0.5, for HDL-C): two SNPs near *CETP*, two SNPs at 8p21.3 near the *LPL* (lipoprotein lipase) gene, two SNPs at 11q12.2 near the *FADS1* and *FADS2* (fatty acid desaturase) genes, and one SNP at 21q22.3 near *FLJ4*17330. The SNP rs7903146 located near *TCF7L2* was associated with T2DM and SNP rs7865146, located <3 kb from the Endoglin (*ENG*) gene and was associated with DBP [118].

The conclusion of this study was that the primary genetic determinants of MetS were the same in Asian Indians compared to other populations despite the higher prevalence of MetS in them. Furthermore, little evidence of a common genetic basis for MetS components was found in the sample studied [118]. No SNPs showed a strong association with the MetS phenotype, although several SNPs were associated with unique traits (e.g., SNPs in/near *CETP, LPL*, and *FADS1-*2 with HDL-C or SNPs in T2DM) [118].

The STAMPEED consortium carried out a GWAS that included seven studies (a total of 22,161 participants of European origin) in which approximately 2.5 million SNPs were analyzed. Patients had MetS or pairs of components of MetS defined according to the NCEP ATPIII criteria [119].

Five SNPs associated with MetS were identified, located within three genes of the *APOA5* cluster (*BUD13, ZNF259,* and *APOA5*), the *LPL* and *CETP* genes. Another 27 SNPs in/near 16 genes were associated with some bivariate combinations of 5 MetS components, including *LPL* (associated with BP-HDL, TG-BP, TG-glucose, HDL-TG, HDL-WC), *CETP* (BP-HDL, HDL-glucose, HDL-TG, and HDL-WC), *APOA5* cluster (TG-BP, TG-glucose, HDL-TG, and WC-TG), *LIPC* (HDL-glucose and WC-HDL), *GCKR cluster,* including *GCKR, ZNF512, CCDC121,* and *C2orf16* genes (WC-TG and TG-BP), *TRIB1* (HDL-TG, TG-BP), *TFAP2B (*WC-glucose), *ABCB11* (HDL-glucose), *LOC100129500* (HDL-TG), *LOC100128354/MTNR1B* (BP-glucose, HDL-glucose, and TG-glucose), and *LOC100129150* (HDL-TG and HDL-WC). The authors concluded that a small part of the covariation between MetS traits can be explained by the pleiotropic effects of the genes involved [119].

The GWAS by Kristiansson et al. [24] included 2637 Finnish patients with MetS (defined according to the International Diabetes Federation, IDF 2005 criteria) and 7927 controls, using ~1.3 million SNPs. In all four cohorts analyzed, a single SNP rs964184 located in the *APOA1/C3/A4/A5* cluster (known to be associated with lipid disorders) was significantly associated with MetS. Other SNPs located in four lipid-associated loci *(APOB, APOA1/C3/A4/A5, LPL,* and *CETP*) were significantly associated with the TG/HDL/WC factor, but none were associated with two or more individually uncorrelated MetS components [24].

Lind et al. [25] performed a GWAS using data from 291,107 individuals from the UK biobank. The authors analyzed the MetS binary trait (NCEP harmonized criteria) and identified 93 independent loci with *p* < 5 × 10^−8^, of which 80 were not identified in previous GWASs of MetS [25]. Most of those loci had previously been associated in other GWASs with one or more of the five components of MetS, but new loci that had not previously been linked to MetS were also identified. The study mainly analyzed genes related to MetS (binary) (*WDR48, KLF14, NAADL1, GADD45G,* and *OR5R1* genes), but also *SNX10* and *C5orf67* genes associated with all five components of MetS in previous studies [120].

The authors identified three loci that were linked to all five MetS components: SNP rs7575523—the closest *LINC0112* gene—rs3936511-intron of *C5orf67,* and rs111970447—intron of the *GIP* gene. The *C5orf67* gene appears to be of interest as a candidate gene for clustering MetS risk factors, being linked to all five MetS components in previous GWASs. Additionally, loci were identified that would be related to the different combinations of four or three components of MetS. Each combination of MetS components showed a unique genetic profile, and the genetic overlap between these combinations was low [120].

The GWASs have revealed an interaction between common genetic factors involved in lipid metabolism disorders (for example, *APOE* (LDL-C), *CETP* (HDL-C), and *LPL* (TG) gene mutations) and genes involved in other metabolic pathways, such as the *GCKR* gene (glucokinase regulatory protein) involved in glucose metabolism. The *PCSK9* gene involved in a form of FH has been intensively studied, with certain variants being associated with reduced plasma LDL-C levels and a reduced risk of CHD. Its identification led to the development of new lipid-lowering drugs—monoclonal antibodies that reduce plasma LDL-C levels by inhibiting *PCSK9* [121].

Other population studies have investigated the impact of low-frequency allelic variants (minor allele frequency 5%) on lipid metabolism. The total variation in lipids caused by common and rare allelic variants was found to be ~40% (heritability), indicating that much of the variation remains unexplained. Animal and human model studies have provided evidence demonstrating that the gut microbiome plays a key role in hepatic and biliary lipid metabolism, as well as cholesterol transport [121].

### 6.1. APOB Gene Polymorphism

Apolipoproteins play an essential role in the transport and metabolism of TG and cholesterol. ApoB is a major protein component of all types of pro-atherogenic lipoproteins, including VLDL, intermediate-density lipoprotein (IDL), and LDL. Elevated plasma ApoB level is an essential biomarker of CVD [69].

Richardson et al. [70] performed a Mendelian randomization study, showing that increased ApoB levels predicted a reduced lifespan associated with a significantly higher risk of CVD [70].

The *APOB* gene polymorphism has been intensively studied in numerous GWASs. The *APOB* RsaI and EcoRI SNP polymorphisms cause amino acid substitution or the loss of restriction sites by affecting specific endonucleases, while other SNPs do not change the amino acid sequence (silent mutations). In the case of the two SNPs, their association with an increased cardiovascular risk has been proven [69]. The *APOB* Xba I polymorphism (rs693) has been associated with a modified lipid profile in many studies [69]. The C and T alleles contribute to the *APOB* rs693 polymorphism. The T allele is a minor allele and is considered a risk allele [69].

Over time, the association between *APOB* rs693 polymorphism and plasma lipid levels has been investigated. Niu et al. [71] reported a significant association between SNP rs693 and increased levels of TC, LDL, and TG [71]. A similar result was obtained by Alves et al. [69] in a study that included 644 elderly Brazilian patients. The authors showed that the homozygous TT genotype was associated with increased plasma levels of TC, total lipids, and LDL compared to homozygous or heterozygous genotypes for the C allele [69].

Although other studies have found divergent results, in the meta-analysis by Kim et al. [72], carriers of the T allele had elevated plasma levels of TC, TG, and LDL [72]. The contradictory results from different studies could be explained by the genetic heterogeneity of the different populations and gene–environment interactions, including different eating habits and lifestyles [72].

APOB-related familial hypercholesterolemia (FH) is the most common inherited autosomal dominant hypercholesterolemia. Karami et al. [73] analyzed the relationship between the polymorphisms rs693 (in exon 26 of *APOB*) and rs515135 (5′ end of *APOB*) (SNPs) in the case of 120 cases of familial hypercholesterolemia and 120 controls belonging to the Iranian population. The authors did not find a significant association between SNP rs515135 and FH but did find a significant association between the C allele of rs693 and elevated plasma cholesterol levels. In addition, the autosomal dominant T allele appears to have a protective role in the onset of the disease [73].

Alghamdi et al. [74] analyzed the relationship between *APOB100* rs693 polymorphism, body mass index (BMI), and the probability of MetS in 141 young females (aged between 18 and 25 years) from Saudi Arabia [74]. The authors reported a significant association between *APOB100* rs693 and elevated plasma TG, TC, and glucose levels in people with MetS compared to the control group. There was no significant association between other genotypes (AA/AG/GG) and lipid parameters, including plasma ApoB100, LDL, HDL levels, and MetS in the control group [74].

It is not yet known how *APOB100* rs693 polymorphism affects lipid profiles [75]. It is possible that there is an association between SNP rs693 with other *APOB* alleles or other nonallelic genes, a phenomenon called linkage disequilibrium [75]. In a study that included patients with and without obstructive sleep apnea, Li et al. [76] reported non-significant associations between SNP rs693 and multiple allelic variants in the *APOB* gene [76]. Components of MetS, including elevated TG and low HDL levels, are key risk factors for obstructive sleep apnea. The authors concluded that SNP rs693 is related to different components of MetS, including plasma LDL and HDL levels, but also with plasma TC, TG, total lipids, and ApoB100 levels [76].

Another intensely studied polymorphism, located in exon 1 of the *APOB* gene, was SNP rs17240441 [69,71]. Two allelic variants in *APOB*, insertion (ins) and deletion (del), contribute to the polymorphism of rs17240441. Cardiovascular risks are attributed to the del-allele. A meta-analysis that included 23 studies reported increased plasma TC, LDL, and ApoB levels in del-allele carriers, compared to non-del allele carriers, while the association was insignificant for plasma TG and HDL levels [71]. The association between SNP rs17240441 and lipid profile was reported in different studies that analyzed both healthy patients and a group of adolescents with essential hypertension (with or without hypercholesterolemia) or patients infected with human deficiency virus (HIV) treated with antiretrovirals [71].

Apolipoprotein A1 (*APOA1*) gene polymorphism is related to plasma HDL-C levels. ApoA1 has a protective effect against atherogenesis, with this role being attributed to its inhibitory effects against platelet aggregation at the level of vascular lesions by prostacyclin stabilization. Furthermore, ApoA1 has an essential role in the reverse transport of cholesterol from peripheral tissues back to the liver by interacting with various receptors. Certain *APOA1* allelic variants, together with dyslipidemia and inflammation, may increase the risk of vascular stiffness in a study by Suparajee et al. [77]. The authors showed an association between the A allele of the *APOA1* rs670 polymorphism and HDL-C levels, high-sensitivity C-reactive protein (hs-CRP), and a lower risk of arterial stiffness in elderly people [77].

### 6.2. APOE Gene Polymorphisms

Apolipoprotein E (ApoE) is a ligand of the LDL receptor (LDLR), and through the LDLR, ApoE is involved in the clearance of VLDL debris and chylomicrons (CM) [78]. Apo E is encoded by the *APOE* gene located on chromosome 19q13.32, closely related to the *APOC-I/C-II* gene complex [14].

In the European population, three *APOE* allelic variants are present (ε2, ε3, and ε4), which determine six genotypes (*APOE2/2, APOE2/3, APOE2/4, APOE3/3, APOE3/4, APOE4/4*), which encode three major apoE isoforms (ApoE2, ApoE3, and ApoE4) [78,79].

Depending on the genotype, the affinity for LDLR varies, which causes significant differences in plasma TC and LDL-C levels.

Some studies have provided evidence regarding the relationship between *APOE* polymorphism and increased risk for CVD and MI. The presence of the ε4 allele is associated with early atherosclerosis, increased mortality, risk of stroke, and MI [78,79].

Heterozygous carriers of the ε4 allele have an 8.3% higher LDL-C level than individuals with the homozygous ε3 genotype; heterozygous individuals carrying the ε2 allele have plasma LDL-C levels 14.2% lower compared to homozygotes for the ε3 allele [78,79].

The risk of death in CAD individuals carrying the ε2 allele was 40% higher than in individuals with the *APOE3/3* genotype or heterozygous ε2 carriers [80].

The rate of death caused by CAD in heterozygous male carriers of the ε4 allele was 1.8 times higher compared to carriers of the other *APOE* allelic variants in the study by Gerdes et al. [81].

The risk for CAD was significantly dependent on the interaction between *APOE* genotype and smoking status in the study by Humphries et al. [82]. This result suggested the possibility that the phenotypic variability of CAD related to the *APOE* genotype was the consequence of the interaction between genotype and environmental factors [82].

### 6.3. APOA1/C3/A4/A5 Haplotype Polymorphism

GWASs have also identified other common SNPs contributing to dyslipidemia located in or near the *APOA1/C3/A4/A5* haplotype.

van de Woestijne et al. [83] showed that carriers of G alleles (minor alleles) for SNP rs964184 located near the *APOA1/C3/A4/A5* haplotype significantly increased TG and ApoB levels associated with low plasma HDL levels [83]. In individuals with the heterozygous genotype for *APOA1* SNP rs964184, BMI was shown to be a predictor for plasma TG levels [83].

Teslovich et al. [84] analyzed more than 100,000 people of European origin and identified a significant association (*p* < 5 × 10^−8^) in the case of more than 95 loci, 59 of which presented for the first time a significant association at the whole genome level with lipid parameters [84]. Recently reported data include SNPs located near known lipid regulators (e.g., *CYP7A1, NPC1L1,* and *SCARB1*) but also numerous loci that had not previously been implicated in lipoprotein metabolism. The 95 identified loci determine both the normal variation of lipid parameters and the appearance of extreme lipid phenotypes in Asians and African Americans. The authors identified three new genes, *GALNT2, PPP1R3B,* and *TTC39B,* associated with an increased risk of coronary atherosclerosis [84].

A meta-analysis of non-European cohorts revealed that in the East Asian cohort, there was a significant association between HDL and TG levels with *APOA1* SNP rs964184; plasma TG level was the only lipid parameter associated with SNP rs964184 in the South Asian cohort. In the case of the African–American cohort, no significant association was found between SNP rs964184 and lipid parameters [84]. The results of other studies conducted on the Chinese population further supported the role of SNP rs964184 in lipid metabolism [85].

Wojczynski et al. [86] conducted a GWAS in European and Amish populations to investigate the role of genetic variants on postprandial TG levels. The authors identified a significant association between SNP rs964184 and postprandial TG levels. This association decreased when initial TG levels were controlled, suggesting that the rs964184 SNP correlates more with initial plasma TG levels than with postprandial TG levels [86].

A similar result was obtained by Alcala-Diaz et al. [87], who reported elevated postprandial TG levels in individuals heterozygous for risk G alleles. In them, a low-fat diet for three years significantly decreased postprandial TG levels, comparable to individuals with homozygous CC genotype, which suggests that SNP rs964184 can be modulated by environmental factors (e.g., diet) or by the gene–environment interaction [87].

The mechanism of how SNP rs964184 affects lipid parameters could be attributed to the localization of the SNP in the 3-UTR region of zinc finger 1 gene (*ZPR1*) [75,87]. The 3-UTR region plays a critical role in regulating transcription, forming mRNA and protein synthesis (translation). The interaction of the *ZPR1* gene promoter with peroxisome proliferator-activated receptor gamma (PPARG) proteins 1 and 2 plays a key role in cholesterol homeostasis. PPARG2 is expressed mainly in adipose tissue, while PPARG1 is expressed in most tissues. The activation of PPARs stimulates the binding of the PPAR-coactivator complex to the promoter region of some genes (e.g., *ZPR1* gene promoter), leading to the activation or inhibition of those genes. The activated genes intervene in the regulation of the plasma cholesterol level by activating the hepatocyte nuclear factor-4 alpha [75,87].

### 6.4. LPL Gene Polymorphism

Lipoprotein lipase (LPL) is an enzyme involved in the metabolism of triglyceride-rich lipoproteins and acts at the level of the vascular endothelium to which it attaches via glycophosphatidylinositol (GPI)-anchored high-density lipoprotein-binding protein 1 (GPIHBP1). LPL plays an important role in lipid metabolism by hydrolyzing circulating triglyceride-rich lipoproteins such as chylomicrons and VLDL [63]. Along with the *LPL* polymorphism, mutations of genes (*APOA5, APOC3,* and *ANGPTL3*) that regulate its endogenous activity can influence CAD susceptibility [88,89].

Angiopoietin-like proteins (ANGPTL) have a similar structure to angiopoietin (ANGPT), which are involved in the metabolism of lipoproteins, especially related to LPL. ANGPTL inhibits the enzymatic activity of LPL and increases the cleavage of LPL, which may lead to an increase in the level of circulating lipids, especially TG [62]. ANGPTL3 and ANGPTL8 are inhibitors of LPL-mediated plasma triglyceride (TG) clearance. The mechanism by which the two proteins interact and regulate LPL activity is not fully known. ANGPTL3 is known to inhibit LPL activity and increase plasma TG level independent of ANGPTL8, whereas ANGPTL8 requires ANGPTL3 expression to inhibit LPL and increase plasma TG level [64]. ANGPTL4 is a strong inhibitor of LPL, causing the decrease of affinity of LPL towards GPIHBP1 and dissociation from it [66].

Lipoprotein lipase deficiency (LPLD) (a monogenic autosomal recessive disease) is caused by homozygous or compound heterozygous mutations in the *LPL* gene. Many studies have reported over time an association between the *LPL* gene polymorphism and susceptibility to CAD/MI, sometimes with contradictory results.

The *LPL* locus can be alternatively occupied by both common non-coding and rare allelic variants. Increased risk of CAD is associated with rare mutations that lead to a loss-of-function (LOF) in the *LPL* gene, while gain-of-function (GOF) mutations in the *LPL* are associated with reduced susceptibility to CAD [88].

Several meta-analyses have suggested that compared to non-carrier individuals, heterozygous individuals carrying the Ser447Ter substitution have a protective lipoprotein profile, while carriers of the Gly188Glu, Asp9Asn, and Asn291Ser substitutions have an increased atherogenic risk [88].

He et al. [90] showed—in a recent meta-analysis—that the *LPL* HindIII and S447X polymorphisms, but not PvuII, could play a protective role against MI. These results require confirmation by further studies that include a larger number of analyzed subjects [90].

In another meta-analysis, Ma et al. [91] identified a significant association between the *LPL* S447X polymorphism and CAD susceptibility in Caucasians. The *LPL* D9N polymorphism was associated with an increased risk of CAD, while the *LPL* HindIII polymorphism had a protective effect. The authors did not identify any significant association between *LPL* N291S and PvuII polymorphisms with CAD risk [91].

Analyzing a cohort of 2708 healthy middle-aged European men, Talmud et al. [92] found that carriers of the Asp9Asn polymorphism who were smokers had a 10.4-fold increased risk for ischemic heart disease (IHD)/CAD compared to non-smokers and non-carriers of the respective allelic variant; smokers but non-carriers of the allele had a 1.6 times higher risk than non-smokers. In the case of the Asn291Ser allelic variant, no significant association with IHD/CAD was found [92].

## 7. The Interaction of Genes Involved in Lipid Metabolism and Atherogenic Dyslipidemia and the Other Components of MetS (Hypertension, Obesity, Insulin Resistance, and Hyperglycemia)

In a study that included 3575 subjects from the cohort Doetinchem, Povel et al. [122] analyzed 373 SNPs of genes involved in lipid and glucose metabolism that have pleiotropic effects in MetS. The prevalence of MetS conditions (hyperglycemia, abdominal obesity, decreased HDL-C level, and hypertension) was measured twice over 6 years. Associations between SNPs and individual MetS features were analyzed by log–linear models. In the case of SNPs linked to several phenotypic manifestations of MetS (*p* < 0.01), the authors investigated whether these associations were independent of each other [122].

Two SNPs, *CETP* Ile405Val (*p* ≤ 0.0001) and *APOE* Cys112Arg (*p* = 0.001), known mainly for their role in lipid metabolism, were associated with low plasma HDL-C levels and abdominal obesity. The association of the two SNPs with HDL-C levels was partially independent of the association with abdominal obesity and vice versa. The authors concluded that these polymorphisms (SNPs) could explain the association between plasma HDL-C levels frequently present in patients with abdominal obesity [122].

The *LPL* Ser447Ter polymorphism is the most intensively studied regarding its association with blood pressure (BP) and risk of hypertension. The results of previous studies have been contradictory, suggesting both positive and negative associations between *LPL* polymorphism and hypertension [123].

Apparently-healthy carriers of the *LPL* Ser447Ter allelic variant had lower values of systolic blood pressure (SBP) and pulse pressure [124,125,126]. In other studies, a correlation between *LPL* Ser447Ter and hypertension was observed in subjects with phenotypic features of MetS [127,128,129]. These contradictory results could be caused by the small size of the analyzed sample, the diverse genetic background, and the different inclusion/exclusion criteria established for each study [75].

Hyperinsulinism and insulin resistance are closely related to several common diseases, including T2DM, CVD, and MetS. Shakhanova et al. [130] analyzed the association between polymorphism of *LPL* (lipoprotein lipase), *ADRB2* (β2-adrenergic receptor), *AGT* (angiotensinogen), and *AGTR1* (angiotensin II type 1 receptor) genes with risk of hyperinsulinism and insulin resistance in a case-control study that included 460 subjects aged 18 to 65 from the Kazakh population. For each patient, plasma glucose, insulin, TC, TG, HDL, LDL, ApoB, and ApoA1 were analyzed.

The quantitative reverse transcription PCR (polymerase chain reaction) method was used to detect *LPL* Ser447Ter, *ADRB2* Gln27Glu, *AGT* Thr174Met, and *AGTR1* A1166C polymorphisms [130].

It is considered hyperinsulinism when the insulin level is elevated >24.9 IU/mL. The authors used the Homeostasis Model of Insulin Resistance Assessment (HOMA) to assess insulin resistance (IR) (HOMA-IR). Subjects were divided into groups with hyperinsulinemia (17 men and 24 women) and normal insulin levels (214 men and 205 women), who were also divided into the insulin resistance group (HOMA-IR > 2.7; 80 men and 105 women) and those without insulin resistance (151 men and 124 women) [130].

The *ADRB2* Gln27Glu (rs1042714), *AGT* Thr174Met (rs4762), and *AGTR1* A1166C (rs5186) polymorphisms were not associated with hyperinsulinism and insulin resistance in the Kazakh population, but carriers of the G allele of *LPL* Ser447Ter (rs328) were associated with a lower risk of hyperinsulinism (*p* = 0.037) [130]. No interactions were identified among the *LPL* Ser447Ter, *ADRB2* Gln27Glu, *AGT* Thr174Met, and *AGTR1* A1166C genes. Thus, the results of this study indicated that this haplotype was not associated with insulin resistance in the analyzed population [130].

Many studies have shown that the prevalence of MetS related to insulin resistance and possibly genetic factors is increased in Asian Indians [131].

Maistry et al. [131] aimed to determine genetic variants associated with MetS in Asian Indians from Durban, South Africa. The study included 999 participants in the Phoenix Lifestyle Project who were clinically and biochemically evaluated for MetS criteria [131].

Four SNPs related to lipid metabolism and the risk for the MetS were selected and analyzed: *APOA5* Q139X (rs121917821), *CETP* (*Taq1B*) (cholesterol ester transfer protein Taq1B) (rs708272), *LPL* Hinf I (rs328), *PON1* (human paraoxonase 1) 192Arg/Gln (rs662), and two SNPs associated with obesity: *LEP* (leptin) 25CAG (rs104894023) and *ADIPOQ* (adiponectin) 45T>G (rs2241766) [131].

The results of this study revealed that the prevalence of MetS was high (49.0%), being more frequently detected in women compared to men (51.0 vs. 42.8%). There was no significant difference in genotype distribution between participants with MetS versus healthy individuals. Men with MetS who had the *ADIPOQ* TG and *PON1* AA allelic variants had lower HDL-C levels (*p* = 0.001) and higher systolic blood pressure (SBP) (*p* = 0.018) [131]. These results suggested that lifestyle was the major determinant of MetS in this ethnic group, and genetic risk might be more related to MetS component risk than to MetS as an entity [131].

Apolipoproteins are involved in the modulation of insulin sensitivity, either directly or indirectly, through their interaction with adiponectin or their antioxidant and anti-inflammatory effects [123,132]. Apolipoproteins are also crucial factors in the development of hypertension through dyslipidemia and their effects on atherosclerosis, inflammation, and endothelial dysfunction [115,133].

ApoA1 has anti-atherogenic effects, and some *APOA1* polymorphisms are associated with MetS: the G-75A with hypertension and C+83T with obesity and with higher levels of glycated hemoglobin [123,132].

ApoB triggers many events that lead to hypertension. Infiltration of apoB-containing lipoproteins in the vascular wall determines inflammatory response and endothelial dysfunction; interactions with proteoglycans initiate atherosclerosis, and consequently, structural vascular anomalies lead to elevation of blood pressure. Han et al. [134] reported apoB level as a useful marker for the development of hypertension independent of abdominal visceral fat and insulin sensitivity [134]. Nayak et al. [135] indicated a significant positive association between ApoB100/ApoA1 ratio and ApoB100 level with systolic and DBP in patients with essential hypertension [135]. Additionally, ApoB levels are strongly associated with the increased risk of newly-onset diabetes and could be a good predictor factor [136].

ApoC3, an important regulator of TG levels, inhibits lipoprotein lipase (LPL) and has proinflammatory effects. Serum level of ApoC3 is negatively associated with insulin sensitivity. *APOC3* SNPs are associated with susceptibility to hypertension: rs4520 in carriers with BMI < 25 kg/m^2^ and rs2854116 or rs2854117 in participants with low physical activity [101].

The *APOE* ε4 allele is one of the most known common variants affecting the risks of some chronic diseases (CVD, Alzheimer’s disease). A meta-analysis investigating the association between *APOE* polymorphisms and hypertension showed that the *APOE* ε4 allele and the genotypes ε3/ε4 and ε4/ε4 are risk factors for hypertension [137].

Conflicting results have been reported regarding T2DM. While some studies have indicated an association between the *APOE* ε4 allele and T2DM [138], Lumsden et al. [139] suggested that the ε4 allele has a potential protective role against obesity, T2DM, and liver disease [139].

Adiponectin is a peptide produced predominantly by adipocytes, with pleiotropic effects on body weight regulation, insulin sensitivity, inflammation, and apoptosis. Serum adiponectin forms large multimeric complexes (trimers, hexamers) and binds to two specific receptors, AdipoR1 and AdipoR2, involved in the activation of the AMPK and PPARα pathways. AdipoRs are expressed in numerous tissues, including the liver, pancreatic β–cells, adipose tissue, muscles, blood vessels, brain, heart, bone, and immune cells [140]. Adiponectin plays a key role in sphingolipid metabolism by activating ceramidase activity and reducing the levels of lipotoxic ceramides [141]. A systematic review by Hafiane et al. [142] confirmed the critical role of adiponectin in promoting ABCA1-dependent cholesterol efflux and increasing HDL synthesis and, consequently, the antiatherogenic effect [142]. An inverse correlation between adiponectin levels and obesity or visceral fat and insulin resistance has been reported [75]. Many studies reported positive effects of dietary interventions and exercise on adiponectin levels [143,144]. Adiponectin is encoded by the *ADIPOQ* gene, located on 3q27.3. Conflicting results have been reported regarding the association between obesity and some *ADIPOQ* polymorphisms (rs1501299, rs17366568, and rs822396). This discrepancy could be attributed to genetic or epigenetic differences between populations. Age could have an important role because studies reported a positive association of rs1501299, rs822396, and rs17366568 with obesity in persons above 38 years old [75,145].

Adiponectin has beneficial effects on glucose metabolism by activating AMPK, which leads to glucose transporter-4 (GLUT-4) translocation to the cell membrane and increases glucose uptake and fatty acid oxidation in skeletal muscle [146]. Many studies reported a strong correlation between serum adiponectin levels and insulin sensitivity. Adiponectin level could be a useful biomarker for the development of T2DM. Furthermore, several SNPs in the *ADIPOQ* have been reported to be associated with the development of insulin resistance and T2DM in different populations: rs2241766 in Japanese, Spanish, Iraqi, Irani, and American Caucasian populations [147,148,149,150], rs266729 in French Caucasian and Chinese populations [151,152], rs17366743 in French Caucasians [151], rs182052 in African Americans and Chinese populations [153,154], rs17300539 in Chinese populations [154], and rs266729 in South Indians [155]. Simeone et al. [156] reported a dominant frameshift deletion in the *ADIPOQ*, resulting in a premature stop codon in a diabetic family with hypoadiponectinemia, hyperceramidemia, and renal disease [156].

Adiponectin has anti-inflammatory effects by stimulating monocyte to M2 macrophage differentiation and suppressing M1 macrophage activation. Serum adiponectin level has been correlated with inflammation severity and disease progression in some autoimmune diseases (rheumatoid arthritis, inflammatory bowel disease) [157].

Adiponectin is involved in AMPK-endothelial nitric oxide synthase activation and COX-2 pathways, with an endothelial protective role and anti-atherosclerotic effects [157]. Iwashima et al. [158] reported that blood pressure is inversely associated with adiponectin level [158]. According to the Fan et al. [159] meta-analysis, hypertension risk is increased by *ADIPOQ* rs2241766 and rs266729 polymorphisms, but rs1501299 may have a protective role in hypertension in the Caucasian population [159].

## 8. Discussions

A full understanding of the role that genetic factors play in the occurrence of MetS and its components (atherogenic dyslipidemia, obesity, insulin resistance, diabetes, and hypertension) is essential in modern medicine. Understanding the complex pathophysiological mechanisms of MetS and identifying all genetic factors involved, as well as translating information provided by new molecular genetic techniques into medical practice, are essential for the management of patients with MetS.

Thus, the new information would allow the development of more effective screening techniques based on accurate risk scores to identify individuals at high risk of MetS and the implementation of early prevention measures, as well as the establishment of new therapeutic targets and gene therapy strategies to treat or prevent disease.

MetS and its components remain an important cause of death worldwide, despite the improvement of treatment and preventive methods [160]. It is known that MetS is correlated with an increased risk for ASCVD and its complications. In this sense, several studies have investigated the association between MetS components as a unified risk factor for CVD. The conclusion was that the CVD risk associated with MetS is not greater than the sum of the risk of its individual components [161]. In addition, the predictive power of MetS as a CVD risk factor for MI, ischemic stroke, and heart failure decreased with advancing age [161,162,163].

The main cause of death in the case of atherogenic dyslipidemia is coronary atherosclerosis, which can have monogenic or multifactorial etiology. GWASs and WES studies have identified numerous loci and risk genes, many of these genetic factors being correlated with a disorder of lipid metabolism and hypertension [13,103,164].

The contribution of genetic factors (hereditability) to the occurrence of MetS and its components has been proven by many family-based studies and Twin studies [13,99]. In the case of MetS, heritability has varied in some studies from 27% [98] to 31.2% [97] and between 16% and 60% for its five components [95]. Despite the success of GWASs, genetic variants that are associated with MetS components typically explain <10% of the variation in the traits across the population, despite heritability measurements of >50% in family-based studies. These aspects suggest that there are definitely genetic factors still unidentified (“missing heritability”). Thus, efforts are being made to identify genetic variants that explain additional variations in MetS traits and their components [164].

Possible causes of “missing heritability” could be rare allelic variants, copy number variation (CNV), microRNA (*miR*), epigenetic regulation mechanisms, different gene-gene interactions (epistasis), or genetic and environmental factors [164].

CNVs typically involve DNA deletions or duplications that can range in size from 50 bp to several megabases. CNV is the most frequent structural variant in the human genome, contributing to genetic heterogeneity, and has been associated with early-onset severe obesity [164].

*miR*s are short segments of transcribed RNA that bind complementary messenger RNA preventing translation and degradation of the primary transcript [165]. Increased expression was found to lead to increased FFAs synthesis and decreased HDL-C efflux, two hallmarks of MetS [164,165].

DNA methylation is an epigenetic modification that influences gene expression. Genome-wide methylation studies have reported differences between T2DM patients and controls [120,166]. Sequencing of genes identified by GWAS identified the presence of rare LOF mutations, especially in individuals with severe phenotypic manifestations of MetS [120,166].

The use of next-generation sequencing techniques (NGS) or WES, as well as gene panels, has become more and more accessible in recent years due to the decrease in the costs of these analyses. NGS techniques allow genotyping of rare mutations, common SNPs, and CNVs [164,165]. Large databases of control sequences are now available (https://gnomad.broadinstitute.org accessed on 25 May 2023), and efforts are ongoing for large-scale sequencing examining not only monogenic diseases but the whole spectrum of complex traits [164,167].

Both in the case of atherogenic dyslipidemia and in the other components of MetS, genetic factors act independently of the environmental factors, and the variable phenotype is the result of their permanent interaction. In addition, possible interactions between different genes (epistasis), epigenetic regulatory mechanisms, as well as interactions between genetic and environmental factors, which GWASs and WES cannot identify, must be taken into account [164].

The marked variability of the heritability of MetS in different studies could also be partially attributed to the ethnicity of the individuals analyzed [13,98,130]. Starting from the evidence regarding the heritability of MetS, but also of its individual components, various studies were carried out that aimed to identify both the genetic determinants for each of the individual components of MetS (dyslipidemia, obesity, hypertension, and DM), but also of common genetic factors, the pleiotropic effect of some involved genes being already known [13,130].

### 8.1. Challenges Related to Future Research in Atherogenic Dyslipidemia and Metabolic Syndrome

Identifying all the factors involved in atherogenic dyslipidemia and the other components of MetS is a real challenge for future research and will certainly contribute to solving this real puzzle represented by the complex genetic etiology of MetS. In the coming years, the integration of information from omics data analysis (genomics, epigenomics, transcriptomics, proteomics, and metabolomics) and their correlation with the phenotypic manifestations (phenom) of MetS will constitute a major challenge. The information provided will contribute to the molecular redefinition of MetS phenotypes and will certainly contribute to the development of targeted therapies [168].

Epidemiological research in recent decades has uncovered a multitude of biomarkers that may be associated with MetS and its components, including the risk of atherogenic dyslipidemia and ASCVD. However, even with strong evidence of their association with the disease, no conclusions could be drawn attesting to the causal relationship between these biomarkers and MetS or its components [168,169].

Mendelian randomization (MR) studies may be able to demonstrate the existence of a causal relationship between a biomarker and MetS or its components if the biomarker is involved in the occurrence of the disease and if the observed association is influenced by unrecognized external factors or the disease itself affects biomarker level [111,169]. Over time, it has been discussed about the opportunities and challenges of MR studies in the case of atherosclerotic disease (especially CAD) associated with atherogenic dyslipidemia, being used several biomarkers involved in both lipid metabolism, inflammation, obesity, DM, and hypertension [111,169]. Analysis of the causative factors of MetS and its components by MR has tremendous potential to identify new therapeutic targets [111,169].

In the future, research based on a large number of patients and the exchange of information between different study groups, as well as the creation of common and accessible databases, will contribute to the elucidation of contradictory results from studies conducted on a small number of cases.

The elucidation of the genetic differences that underlie the different susceptibility to MetS and its components and the particular response to treatment in individuals of different ethnicities will allow the development of targeted and personalized therapies [13].

### 8.2. The Importance of GWASs and Polygenic Risk Scores (PRS) for the Prevention of Atherogenic Dyslipidemia and MetS

The knowledge of the genetic architecture of atherogenic dyslipidemia and other components of MetS has clinical applications related to the identification of new therapeutic targets and the development of new gene therapies, but also to estimate the risk for CVD [13,103].

GWASs have shown that common complex diseases have a multifactorial (polygenic) etiology and have allowed the identification of genetic variants associated with these diseases. These allelic variants can be included in a polygenic risk score (PRS) to detect part of the individual’s susceptibility to that disease [170].

Atherogenic dyslipidemia and MetS are considered important risk factors for ASCVD. Thus, many studies have demonstrated that the risk for ASCVD and CAD can be assessed by the PRSs using common genetic variants [170]. PRSs have been developed to assess the risk of CAD, which now include millions of SNPs that have been identified through GWASs. Many genetic variations of small effects contribute to an individual’s susceptibility to ASCVD/CAD. The PRS prediction quantifies the contributing effects in a score and estimates whether the tested individual is at high and medium risk for ASCVD [170,171]. A PRS is calculated as the sum of a number of genomic variants weighted to estimate their effect, which has been determined by GWASs [170,171].

Initially, most of the PRSs for ASCVD and CAD were made for the European population, not being able to be used in populations of other ethnic origins. Along with the identification of new genetic factors of ASCVD, the elucidation of genetic differences related to ethnic origin has been a challenge for research both in the case of dyslipidemia and other components of MetS [170]. The predictive power of PRSs for ASCVD and CAD was improved by including evidence of association, linkage disequilibrium, pleiotropy, and trans-ancestry genetic correlation that allowed the use of PRSs in populations of different ethnicities [170,171].

The use of PRS to identify people at increased risk of ASCVD and CAD would allow the establishment of effective preventive measures through lifestyle modification and early therapy with lipid-lowering drugs (statins). The results of some clinical studies have proven the effectiveness of early therapy aimed at lowering LDL-C in the case of people with a high PRS for ASCVD and CAD [170,172].

Yaghootkar et al. [173] performed a composite risk score of 11 SNPs previously reported to be associated with insulin resistance [173]. The score generated by them was significantly associated with higher TG levels, lower HDL, hepatic steatosis, reduced adiponectin, lower BMI, and increased risk of T2DM and CAD. The authors suggested that patients with an unusual combination of risk alleles represent a polygenic “lipodystrophy-like” phenotype [173].

### 8.3. Prophylactic Measures in Families at High Risk for Atherogenic Dyslipidemia and MetS

Patients with increased genetic risk for dyslipidemia and MetS require personalized prevention measures and treatment strategies. In their case, effective prevention methods have been shown to include avoiding traditional cardiovascular risk factors (CRFs) and adopting a healthy lifestyle with a balanced diet, avoiding excess fat consumption and smoking, and regular physical exercise, all of which are correlated with the decreasing of the severity of the disease manifestations and the occurrence of acute complications such as MI [174]. The use of guidelines for risk assessment as well as the approach of personalized prophylactic strategies based on family risk represent current practice in the management of patients/families at high risk of MetS and CVD [171].

There is evidence showing that the gut microbiota is one of the most important pathogenic factors in MetS [175]. MetS itself is caused by the interaction between intrinsic factors of the host (genetic background and gut microbiota) and extrinsic factors (eating habits and lifestyle). MetS is often accompanied by an imbalance of the intestinal microbiota, which induces a reduced inflammatory response of the body due to the destruction of the intestinal barrier, which causes insulin resistance (through metabolites that affect the metabolism of the host) and the release of hormones, forming a vicious circle that favors continuous progress of MetS [175].

Thus, the gut microbiota could be a potential target for the treatment of MetS. Further research is needed to understand how the gut microbiota can be manipulated to be used in the prevention and treatment of MetS. In the future, these discoveries will probably lead to the development of new therapeutic strategies in MetS [176].

### 8.4. Genetic Counseling in Families at High Risk for Atherogenic Dyslipidemia and MetS

Calculating the risk of recurrence and giving genetic counseling in families at high risk of dyslipidemia and MetS is conducted by taking into account the genetic risk factors present in the family. In affected families, the risk of recurrence is different depending on the etiology; in some cases, a monogenic mutation is detected (especially in the case of ASCVD and CAD), while other cases have a complex, multifactorial etiology. Genetic counseling for patients at increased risk of dyslipidemia and MetS should include, in addition to a detailed physical examination, family history, and pedigree analysis (which may provide data on monogenic inheritance or the existence of other affected family members), personal medical history, lifestyle, diet, and medications used [176,177].

Starting from the fact that most cases of MetS have a multifactorial etiology, assessing the risk of recurrence and giving the correct genetic counseling can be difficult, considering the large number of loci and genes that can intervene, as well as the interaction between different genes (epistasis) or the interaction of genetic and environmental factors [176,177].

The main limitation of our study, in which we proposed a detailed analysis of data from the literature regarding the genetic factors of atherogenic dyslipidemia, was caused by the insufficient data in the literature and the fact that many of the genetic factors involved are not yet identified (“missing heritability”) [164,178].

This aspect could be correlated both with the genetic heterogeneity detected in the case of atherogenic dyslipidemia and the other components of MetS, as well as with the insufficient investigation of the interactions between genetic factors (epistasis) or the interaction between genotype and environmental factors [164,178].

In addition, the interactions between the genetic factors determining atherogenic dyslipidemia with the genes involved in the occurrence of the other components of MetS or with environmental factors were not discussed in detail.

The study of these interactions could elucidate the complex pathophysiological mechanisms of MetS in the future. Deciphering the genetic architecture of atherogenic dyslipidemia and MetS components remains a topic of major interest due to the complexity of the genetic and non-genetic factors involved, as well as the possible interactions between them.

Probably, in the future, new biomarkers will be identified, which, together with the use of PRSs, will improve the prediction of risk for MetS and its components.

Additional research will be needed to confirm the results of small studies, as well as the continuation of studies targeting ethnic differences both in terms of susceptibility to MetS and response to treatment.

In the case of atherogenic dyslipidemia and MetS, we cannot yet speak of personalized medicine, and we do not know how the detected genetic polymorphisms (genotype) could influence the response to treatment.

Most likely, in the future, molecular testing to detect genetic susceptibility in the case of atherogenic dyslipidemia and MetS will be routinely used in medical practice, the goal being to take effective prophylactic measures, thus preventing severe complications.

## 9. Conclusions

With the development of molecular genetics techniques, important progress has been made in deciphering the complex etiology of atherogenic dyslipidemia and the other components of MetS. Although global mortality remains high, the main benefit has been the identification of new therapeutic targets and the development of innovative therapies. GWASs provided information that allowed both the deciphering of the complex molecular mechanisms of MetS and the identification of new genetic factors, which, together with those already known, increased heritability both for MetS and for its individual components, including atherogenic dyslipidemia.

The identification of new genetic factors in the future could explain the “missing heritability” without ignoring the fact that the interaction between different genes or between genetic and environmental factors could determine the phenotypic variability of MetS.

Furthermore, the identification of new genetic factors with a protective role against atherogenic dyslipidemia and ASCVD could constitute an important objective of future research.

Current use of the PRSs could improve risk prediction for MetS and its components, making it possible to personalize both prevention and treatment for each patient in high-risk families.

## Figures and Tables

**Figure 1 diagnostics-13-02348-f001:**
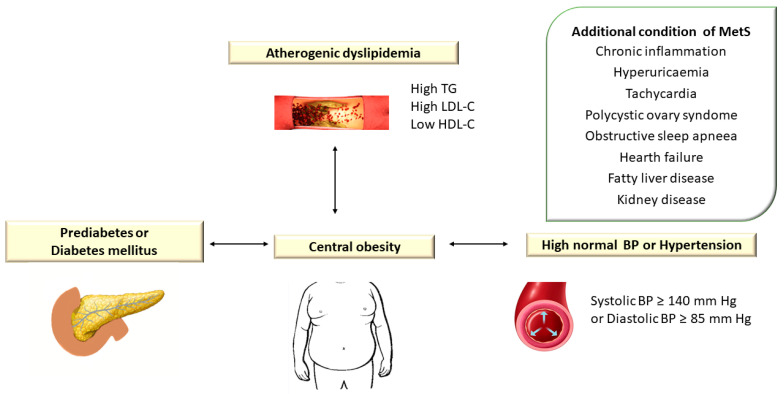
Atherogenic Dyslipidemia Associated with Other Metabolic Disorders in Metabolic Syndrome. TG: triglyceride; LDL-C: low-density lipoprotein cholesterol; HDL-C: high-density lipoprotein cholesterol; BP: blood pressure. In the European guidelines, hypertension is defined as blood pressure (BP) ≥ 140/90 mm Hg [5], while the American guidelines consider values of TA ≥ 130/80 mm Hg [6].

**Figure 2 diagnostics-13-02348-f002:**
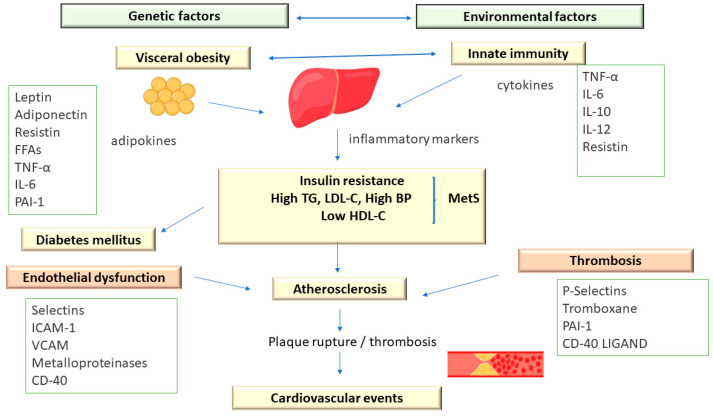
The Pathophysiology of Metabolic Syndrome.

**Figure 3 diagnostics-13-02348-f003:**
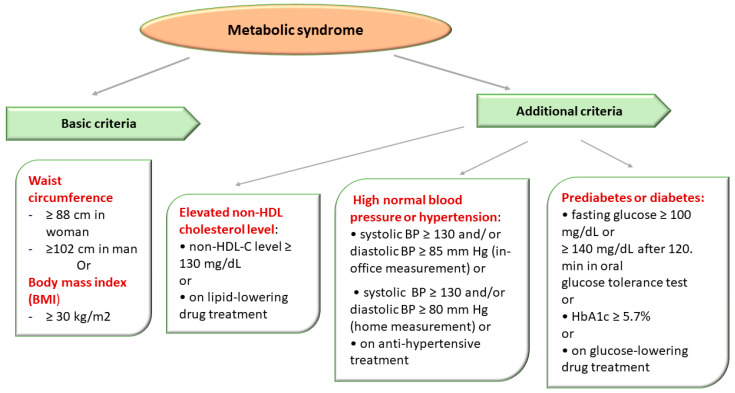
Metabolic Syndrome Diagnostic Criteria Defined in Accordance with The National Cholesterol Education Program Adult Treatment Panel III (NCEP/ATPIII criteria) [2,95].

**Table 1 diagnostics-13-02348-t001:** Genetic Etiology of Atherogenic Dyslipidemia.

Gene(s)	Locus/ Chromosome	Disease	Biochemical Changes	Method CGS/GWAS/LA	Polymorphism/Allelic Variant/ SNPs	References
**Monogenic lipid disorders**						
Genetic Causes of Elevated Plasma LDL-Cholesterol Level: Familial Hypercholesterolemia type 1 (FH)						
*LDLR*	19p13.2	FH	↑ LDL-C	CGS/GWAS		[14,15,16,17,18,19]
*APOB*	2p24.1	FHCL2/FDB	↑ LDL-C	CGS/GWAS		[14,20,21,22,23,24,25,26,27]
*PCSK9*	1p34.1-p32	HCHOLA3	↑ LDL-C	CGS/GWAS		[14,28,29,30]
*LDLRAP1*	1p34-p35	ARH	↑ LDL-C	CGS		[14,31,32,33]
Genetic Causes of Low Plasma HDL-Cholesterol Level						
*APOAI*	11q23.3	Apo AI deficiency, Apo-A1, and apo C-III combined deficiency	↓ HDL-C	CGS		[14,34,35,36,37,38,39,40]
*ABCA1*	9q31.1	TGD	↓ HDL-C	CGS/GWAS		[14,41,42]
*LCAT*	16q22.1	LCAT deficiency	↓ HDL-C	CGS/GWAS		[14,43,44,45]
Genetic Causes of Hypertriglyceridemia						
*LPL*	8p21.3	LPL deficiency/CHLF	↑ TG, ↓ LDL-C, ↓ HDL-C	CGS		[14,46,47,48,49]
*APOC2*	19q13.2	HLIb	↑ TG	CGS		[14,46]
*ABCG5, ABCG8*	2p21	STSL	↑ plant sterols	CGS		[14,50,51]
*GPD1*	12q13.12	HTGTI	↑ TG	CGS		[14,46]
*AGPAT2*, *BSCL2*, *CAV1*, *CAVIN1*	9q34, 11q13, 7q31.2, 17q21.2	CGLS	↑ TG	CGS		[14,46,52]
*PPARG*, *LMNA*	3p25.2, 1q22	FPLD3, FPLD2	↑ TG, ↓ LDL-C, ↓ HDL-C	CGS		[14,52]
*APOE*	19q13.32	FDBL	↑TG, ↑TC, ↓ HDL-C			[14,46]
*Familial Combined Hyperlipidemia and Familial Hypertriglyceridemia*						
*APOA1/C3/A4/A5*	11p14.1-q12.1, 1q21-23, 16q22-24.1	FCHL	↑ VLDL, ↑ LDL-C, ↑ ApoB, ↑ TG	LA/GWAS		[14,53,54,55,56,57,58,59,60,61]
*APOA5*	11p14.1-q12.1, 15q11.2-q13.1, 8q11-q13	FHTG	↑ TG	LA/GWAS		[14,54,61,62,63,64]
*Atherosclerosis Susceptibility/Atherogenic Lipoprotein Phenotype*						
*ATHS*	19p13.3-p13.2	ATHS/ALP	↑ LDL-C ↑ TG, ↓ HDL-C	LA		[14,65,66,67,68]
** *Polygenic lipid disorder* **						
*APOB*	2p24.1			GWAS	rs69, rs17240441	[69,70,71,72,73,74,75,76,77]
*APOE*	19q13.32			GWAS		[78,79,80,81,82]
*APOA1/C3/A4/A5*	11p14.1-q12.1, 1q21-23, 16q22-24.1			GWAS	rs964184	[75,83,84,85,86,87]
LPL	8p21.3				Gly188Glu, Asp9Asn, Asn291Ser substitutions	[62,63,64,66,88,89,90,91,92]

HDL-C: high-density lipoprotein cholesterol; LDL-C: low-density lipoprotein cholesterol; TG: triglyceride; CGS: candidate gene-based association study; GWAS: genome-wide association study; LA: genetic linkage analysis; WES: whole exome sequencing; FH: familial hypercholesterolemia; FDB: familial defective apolipoprotein B-100; FDBL: familial dysbetalipoproteinemia; HTGTI: hypertriglyceridemia, transient infantile; CGLS: monogenic congenital lipodystrophy (CGL) syndromes; FHCL2: hypercholesterolemia, familial, 2; HCHOLA3: hypercholesterolemia, autosomal dominant, 3; ARH: autosomal recessive hypercholesterolemia; TGD: Tangier disease; HLIb: hyperlipoproteinemia type Ib; LPL deficiency: lipoprotein lipase deficiency; CHLF: combined hyperlipidemia, familial; STSL: sitosterolemia; FCHL: familial combined hyperlipidemia; FHTG: familial hypertriglyceridemia; VLDL: very-low-density lipoprotein; APOB: apolipoprotein B; atherosclerosis susceptibility (ATHS)/atherogenic lipoprotein phenotype (ALP).

## Data Availability

Not applicable.

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
