# Peer review of "Current Data and New Insights into the Genetic Factors of Atherogenic Dyslipidemia Associated with Metabolic Syndrome"

_diagnostics, 2023, doi:10.3390/diagnostics13142348_

Round 1
Reviewer 1 Report
This review comprehensively introduces the influence of genetic factors on lipid metabolism disorders associated with the metabolic syndrome. It effectively presents current findings from published literature on the development of lipid metabolism disorders and their impact on the metabolic syndrome. The review also acknowledges the limitations of the approaches used in the referenced and discusses the potential clinical applications of the information presented. However, there are some areas that can be improved in terms of organization and sentence structure:
1. To enhance the cohesiveness of the review, it is advisable to avoid sporadic paragraphs and strive for a clearer overall picture. This can be achieved by connecting the explanations more seamlessly and ensuring that the main argument is evident to the audience.
2. Long sentences should be consolidated to improve readability. Some sentences in the review span multiple lines, which can confuse readers. Breaking them down into shorter, more digestible sentences will help readers follow the content more easily.
3. Consider the necessity of each section in the review. The length of the review can be overwhelming, and including trivial details may burden the reader. Evaluating the relevance and significance of each section can help streamline the content.
4. To enhance accessibility, it is recommended to use colloquial terms when possible. While the review contains numerous scientific terms, incorporating more layman's language can make it more comprehensible to a general audience.
Author Response
Greetings, editors and referees.
We appreciate you taking the time to review our article, and we are honored. Your suggestions were considered, and the necessary adjustments were made.
- Your observation about the fact that our article is very extensive is pertinent. However, we consider that this is due to the complexity of the subject addressed and the large number of genetic factors involved in the occurrence of atherogenic dyslipidemia, the major component of the metabolic syndrome. We cannot omit any of these causes and also not discuss the existence of genes with a pleiotropic effect, also involved in the other components of MetS. Also, we cannot omit the interactions between different genetic factors or between them and environmental factors. We could not omit the importance of using PRS in detecting people with increased susceptibility to dyslipidemia and MetS, as well as the problems related to providing genetic counseling in families at risk.
- Since our paper focuses on the role of genetic factors involved in the occurrence of atherogenic dyslipidemia, the use of specialized terms (related to the nomenclature of genes, their location on chromosomes, the way of writing genetic variants / polymorphisms) is imperatively necessary, being correlated with maintaining scientific rigor in the presentation of information.
Thank you!
Reviewer 2 Report
This is a comprehensive overview of the current knowledge of genetic factors in DM-associated atherogenic dyslipidemia.
I have only minor suggestions:
The authors should underline that risk factors, if clustered in MetS, pose much higher cardiometabolic risk than their simple sum
the authors mention in detail the prevalence of MetS, but not the prevalence of hypertriacylglycerolemia and low HDL-C. Moreover, the comparison between the prevalence in the EU and US is misleading since data sources differ by a decade.
The authors repeatedly mention the lack of standard MetS definition. I would consider the consensus statement from 2009 as a guideline - cut-offs, except for the waist circumference (eg., only the American norm is listed in Fig.3), are identical, and 3/5 components are generally used.
To my knowledge, the MDPI requires reporting of data in SI units (lipids, HbA1C). The authors mix the conventional units with SI throughout the paper.
there are several typo mistakes (e.g., kydneys) sometimes the abbreviations are not spelled-out, interleukines are generally abbreviated IL, not ILN, etc.
Author Response
Greetings, editors and referees.
We appreciate you taking the time to review our article, and we are honored. Your suggestions were considered, and the necessary adjustments were made.
- At your recommendation, we have updated the data on the prevalence of MetS, but we still maintain that the information related to the prevalence of MetS in Europe and the United States took into account approximately similar periods of time.
- For example, for the prevalence of MetS in Europe, I mentioned the study by Scuteri et al - 2015.
- Scuteri, A.; Laurent, S.; Cucca, F.; Cockcroft, J.; Cunha, P. G.; Manas, L. R., et al., Metabolic syndrome across Europe: different clusters of risk factors. Eur J Prev Cardiol 2015, 22 (4), 486-91. doi: 10.1177/2047487314525529.
- Prevalence of Mets in the United States - period 2011-2018
- Ford, E. S.; Giles, W. H.; Dietz, W. H., Prevalence of the metabolic syndrome among US adults: findings from the third National Health and Nutrition Examination Survey. JAMA 2002, 287 (3), 356-9. doi: 10.1001/jama.287.3.356.
- Liang, X.; Or, B.; Tsoi, M. F.; Cheung, C. L.; Cheung, B. M. Y., Prevalence of metabolic syndrome in the United States National Health and Nutrition Examination Survey 2011-18. Postgrad Med J 2023. doi: 10.1093/postmj/qgad008.
- We also took into account your other suggestions, all the changes made being marked in the text.
Thank you,
Dr Tarca / Authors